# New Detection Paradigms to Improve Wireless Sensor Network Performance under Jamming Attacks

**DOI:** 10.3390/s19112489

**Published:** 2019-05-31

**Authors:** Carolina Del-Valle-Soto, Carlos Mex-Perera, Ivan Aldaya, Fernando Lezama, Juan Arturo Nolazco-Flores, Raul Monroy

**Affiliations:** 1Facultad de Ingeniería, Universidad Panamericana, Álvaro del Portillo 49, Zapopan, Jalisco 45010, Mexico; 2ITAM, Rio Hondo 1, Ciudad de México 01080, Mexico; carlos.mex@itam.mx; 3Campus Sao Joao da Boa Vista, Sao Paulo State University, 505 Jardim das Flores, SP., Brazil; ivan.aldaya@sjbv.unesp.br; 4Research Group on Intelligent Engineering and Computing for Advanced Innovation and Development (GECAD), Polytechnic of Porto (IPP), 4000-045 Porto, Portugal; flzcl@isep.ipp.pt; 5Computing Department School of Engineering and Science, Tecnológico de Monterrey, Campus Puebla, Vía Atlixcáyotl No. 5718, Reserva Territorial Atlixcáyotl, Puebla Pue. 72453, Mexico; jnolazco@tec.mx; 6Tecnologico de Monterrey, Escuela de Ingeniería y Ciencias, Carr. al lago de Guadalupe Km 3.5, Margarita M. de Juárez, Atizapán, EdoMex 54766, Mexico; raulm@tec.mx

**Keywords:** detection and mitigation jamming, performance metrics, wireless sensor networks

## Abstract

In this work, two new self-tuning collaborative-based mechanisms for jamming detection are proposed. These techniques are named (i) Connected Mechanism and (ii) Extended Mechanism. The first one detects jamming by comparing the performance parameters with respect to directly connected neighbors by interchanging packets with performance metric information, whereas the latter, jamming detection relays comparing defined zones of nodes related with a collector node, and using information of this collector detects a possible affected zone. The effectiveness of these techniques were tested in simulated environment of a quadrangular grid of 7 × 7, each node delivering 10 packets/sec, and defining as collector node, the one in the lower left corner of the grid. The jammer node is sending packets under reactive jamming. The mechanism was implemented and tested in AODV (Ad hoc On Demand Distance Vector), DSR (Dynamic Source Routing), and MPH (Multi-Parent Hierarchical), named AODV-M, DSR-M and MPH-M, respectively. Results reveal that the proposed techniques increase the accurate of the detected zone, reducing the detection of the affected zone up to 15% for AODV-M and DSR-M and up to 4% using the MPH-M protocol.

## 1. Introduction

Wireless Sensor Networks (WSNs) are made up of hundreds, or even thousands, of sensor nodes with limited communication and calculation capabilities that can cooperate to perform complex tasks such as monitoring or control [1,2,3]. Due to their distributed nature and open access to the wireless medium, WSNs may encounter intermittent communication issues and are extremely vulnerable to different kinds of attacks and security threats [4]. For instance, these networks may be subject to jamming attacks, which deactivate or saturate the resources of the system. This type of attack aims to reduce the network capabilities for transporting information. For achieving such goal, the jammer node generates an interference. If the jammer succeeds then the MAC layer of the wireless devices under the attack may produce packet retransmissions, increasing the network traffic.

In particular, in the event of jamming, the network performance decreases noticeably [5]. A key element of counteracting a jammer node is its detection and localization, being the centralization of the attacker a fundamental step [6]. The optimization of security infrastructures in terms of resources (energy, computing time, overhead and others), provides a useful perspective when facing an attack, minimizing its influence on network performance. A jamming attack is intended to generate interference and it may be difficult to be detected because other types of communication interference can also occur. The jamming detection problem, therefore, consists of segregating the behavior from the wireless environment from the network topology. An abrupt network performance decrease, alongside with constant topology reconfiguration, often indicates a possible attack. Currently, jamming detection techniques [7] are mainly based on basic signal-to-noise ratio deterioration measurements [8], Received Signal Strength Indicator (RSSI), or other metrics such as packet loss or packet delivery ratio [9].

In WSNs, jamming detection techniques must be adapted to the particular nature of these networks and their routing protocols [10]. Among them, reactive routing protocols [11] are broadly adopted in WSNs due to their small packet overhead on the network and the fact that new nodes entries are created only when deemed necessary, which reduces the required number of messages between neighbor nodes. Consequently, the source node requires to find a route before sending a message leading to processing time even before it starts. Ad hoc On-demand Distance Vector (AODV) [12] and Dynamic Source Routing (DSR) [13] are two well-known reactive protocols. AODV uses the routing tables of the nodes to find the route towards to a given destination. The new routes are found by a reactivity mechanism, in which the needed route is asked to the neighboring nodes. In this way, the number of hops towards the requested path is found. Before the process of discovery of any route starts, this table is conformed by neighbor nodes and new destinations are only introduced when a node needs to communicate with another node and the table does not contain a route. Thus, a process to discover the possible routes to the destination is launched. In this process, each node uses a counter or sequence number (i.e., an integer value) that is incremented each time before generating a control message. On the other hand, the DSR protocol follows an origin routing scheme which determines the route of each packet just before sending it. Besides, in this algorithm, the sent packets between nodes must include a header that determines the identifiers of the nodes visited along the path. Fortunately, DSR does not require persistent control messages, which reduce the overhead on the network. Moreover, we introduce a hybrid protocol (between proactive and reactive nature), named MPH (Multi-Parent Hierarchical). The MPH protocol, proposed in [14], creates a hierarchical network logical topology where the hierarchy of the nodes is given by their location level. It works like a hierarchical tree: nodes establish parent and child links that constitute the possible routes. Node hierarchies are used to establish links among parents and children based on the coverage radius that depends on the transmission power. As a result, a node can share both the children and the parents with another node belonging to the same hierarchical level. MPH takes advantage of the proactive route maintenance, allowing more than one route per node, which makes it versatile and adaptable to different topologies.

In this work, two new collaborative mechanisms for jamming detection, based on cooperation by receiving feedback from other network nodes, are proposed. The two proposed techniques are improved versions of the method published in [15] with a free training characteristic, i.e., both techniques do not require a previous learning step. These techniques are named (i) Connected Mechanism and (ii) Extended Mechanism. The first one is based on the comparison of performance parameters with respect to directly connected neighbors by interchanging packets with performance metric information, whereas the latter relays The second one is based on the comparison of the parameters known by a collector node, that establishes differences between them and detect a possible affected zone. A notorious advantage of the these two detection techniques proposed in this work concerning the training technique from [15] is that the implementation is faster because there is no previous learning stage. Another advantage is that if the attacker is active during the learning stage, the reference values of this step will be erroneous parameters and the system will not be able to identify the presence of the jammer.

Here, we restate the novel contribution that a sensor network can be defended more efficiently by taking advantage of the combined knowledge of the sensor nodes and the collector node. The two proposed jamming detection mechanisms work by identifying and tagging the nodes under attack to circumvent the affected zone. The detection mechanism is based on a novel persistence parameter that gives a value in the neighbor tables for each node. Besides, the mechanism can be applied in centralized (**Extended Mechanism**) or distributed (**Connected Mechanism**) mode.

The performance of the proposed approaches is analyzed through computational simulation that analyze the effect of the proximity of the nodes and how a malicious node affects it, thus considering the possibility of false positives resulting from the possible confusion between an affected node and interference in a reactive jamming scenario. Reactive jamming identification is usually based on finding abnormal system behavior and, therefore, it requires the system to be analyzed in the absence of jamming. Also, the effectiveness of the proposed mechanism is contrasted with the mentioned two well-known reactive routing protocols, namely AODV [12] and DSR [16], and one proactive routing protocol, MPH [17]. Results reveal that the proposed technique can reduce the detection of the affected zone to a 15% of the total area identified using AODV and SDR protocols (i.e., reactive protocols), and up to 4% using the MPH protocol (i.e., a proactive protocol). Besides, the performance was assessed in terms of different metrics, namely (i) number of retransmissions and, (ii) retries to listening to the channel, (iii) the end-to-end delay, (iv) resilience, and (v) energy consumption during the attack, showing the effectiveness of the proposed approach.

The rest of the paper is organized as follows: Section 2 introduces the related work on the jamming detection problem. Section 3 proposes the analysis of detection mechanism. Section 4 presents the results and describes the performance metrics and energy consumption under routing protocols and a jamming attack. Finally, conclusions are given in Section 5.

## 2. Related Work on the Jamming Detection Problem

Sensor networks are typically deployed in unattended and wild environments, with a shared, unprotected, wireless communication channel. Besides, the reduced computing capacity of a nodes, its strict energy management, and the complexity of the network, might result in a particularly highly vulnerable system. In addition, attackers usually have direct physical access to the hardware of each node, which provides them with many possible ways to attack either physical devices or the network. For this reason, it is necessary to introduce safety requirements in the specification of sensor networks, while improving the mechanisms used to prevent attacks.

Jamming is considered one of the most harmful attacks in threats to WSNs. A jamming attack can limit the communication capabilities of a WSN by degrading the system performance interfering with signals using certain interfering devices [18]. For instance, the proposal in [19] is based on a jamming detection algorithm that employs honeynodes and response mechanism concepts. Each node cooperates with the transmitter to detect any jamming. The authors analyze network performance metrics such as: duration of jamming attack, control message overhead, and the number of channel re-configurations, as well as one of the metrics they use in their simulations is the Packet Delivery Ratio (PDR).

To disable a node or set of nodes in a WSN, the jammer node transmits a radio-frequency signal that interferes with the legitimate signals used by the other sensor nodes. Lou et al., in [20], mention the importance of the transmission state in the links under jamming and how it influences the routing protocol. In [21], Sheikholeslami et al. analyze the system performance from the network layer through routing. The authors analyze a constant jamming, while in our work we study a reactive jamming that represents the worst-case scenario and the most aggressive attack. In [22], Mustafa et al. study on-demand routing protocols and the presence of one or more jammers and develop a multi-path selection algorithm. The authors test their algorithm with a modified version of the AODV protocol. In the work of Ghaderi et al. [23], they discuss address the problem of routing link selection based on the jamming parameters that is measured on a low energy consumption algorithm. Nevertheless, they do not isolate nodes under attack and, when jamming conditions change, the selected route could be inside of the affected zone. The difference of such work with respect to ours is that they find the best route to a destination, while we isolate the affected area so that the protocol re-configures the valid routes and the information arrives reliably.

In [24], Pan et al. study problems related to energy depletion in hierarchical WSNs models. The authors propose an energy-saving oriented model in which a relay node is used to regulate packet traffic and energy consumption in intermediate nodes. Data gathering, another issue presented in hierarchical networks, is analyzed in [25] under a predefined transmission path scenario. Since a predefined transmission path favors the creation of clusters, a dynamic routing approach is proposed in [26] to balance the traffic in the network and decrease the energy consumption in the nodes. A new performance metric, called energy consumption density, is established to validate the effectiveness of their approach. Against that background, the proposed jamming detection technique is implemented in a hierarchical model of WSNs, which facilitates packet sending and decreases network overhead. Besides, most of the studies use energy-related metrics to analyze the performance of long-lasting and difficult access networks such as WSNs. In contrast, we also introduce five performance metrics to enhance the accuracy in the detection of a possible attacker. The additional metrics proposed in this work are (i) retransmissions, (ii) Carrier Sense Multiple Access (CSMA) retries, (iii) resilience, (iv) delay and (v) valid routes.

Navda et al. [27] implement jamming detection algorithms related to the choice made by the attack strategy of the jammer node. Li et al., in [9], implement a scheme based on a gradient-descent algorithm which evaluates the next nodes to the jammer node.

So here it is very important to apply a suitable technique against a malicious attack is important by depleting the attacker energy, limiting the effects of the jamming in the network. Moreover, in the paper by Wood et al. [28], a packet is used for informing the neighbor nodes that there is a jamming attack in progress. In Wood’s work et al., the notification message initiates a mapping process, which requires the exchange of the information using BUILD messages. The work cited in [29] evaluate different critical scenarios in which the network is under attack and performance behavior is analyzed to detect the jamming. Analyzing the behavior of multi-hop wireless networks, we find that by the wireless nature in which they are located and by the number of nodes that communicate, these networks are prone to suffer vulnerabilities. In [30], detection mechanisms are examined in proactive multi-path routing and the authors study a metric called degree of interference activity, which measures the jamming that is occurring in an area of the network. This perspective presents a good complement technique because the precise delimitation of the affected area is the first tangible step for the mitigation of an attack.

### Proposed Method and Contributions

Our work focuses on finding an alternative where the network can defend itself, without the need for techniques that use large amount of resources, such as cryptography. In our analysis, we emphasize the study of one of the most aggressive cases of jamming (i.e., the reactive), and we propose two techniques inherent to the network so that the nodes, taking advantage of the communication capabilities of a sensor network, can defend against an attacker and counteract its effect. Our proposal incorporates the information of the jammed nodes to the tables of the neighbor nodes so that the self-configuring mechanisms of the routing protocols avoid the jammed area. Moreover, in this work we also present an algorithm for measuring thresholds of performance metrics to note the level of affectation of a node. We intend to cover the best of the previously studied proposals in order to test protocols widely known in sensor networks (proactive and reactive protocols) by modifying their valid routes so that it can guarantee a reliable delivery of information and isolate a possible attacking node from the other nodes in the network. In other words, it makes an accurate detection of the affected area based on more reliable performance metrics.

Specifically, the contribution of this work to the state of the art are as follows: (1) We study a reactive jamming that represents the worst-case scenario and the most aggressive attack. (2) We also present an algorithm for measuring thresholds of performance metrics to note the level of affectation of a node. (3) We add a parameter in the routing tables and a control packet for AODV and DSR protocols. (4) In contrast to other works that find the best route to a destination, in this work we isolate the affected area so that the protocol re-configures the valid routes and the information arrives reliably.

## 3. Analysis of Detection Mechanism

We propose two mechanisms (Extended and Connected) that are inserted in the routing protocol, which presents the following modifications to the protocol. One of the advantages of the Extended Mechanism is that it provides a global view of the complete network to the coordinator node. However, the main disadvantage is that this node requires the performance parameter for each node which results in an undesired overhead. The collector node groups nodes with similar behavior which consists of the distance, the amount of retransmitted traffic and average energy. Thus, the Extended Mechanism establishes the comparison parameters based on a set of nodes with similar features. Since the collector node already has all the information, it can easily spot when a node reports different thresholds than those established in its group. Consequently, an alert is generated, and the node is marked as an affected node. On the other hand, in the Connected Mechanism, each node receives information with the performance parameters of its neighbors directly connected via the HELLO packet, then this is the only reference it must know if the thresholds are within the normal conditions. This mechanism generates less overhead, but the reference is less reliable than the perspective of a conglomerate of nodes.

There is a persistence parameter that is bound to the output of the neighbor’s tables of nodes. The persistence parameter is located as a flag in the neighbor table of nodes. In case a node presents alterations in its performance metrics so that its neighbors can eliminate it as a valid route from their routing tables, there is a control packet that is sent to the neighbors when these performance metrics have an abnormal behavior (different values from their numbers under normal conditions). These metrics are packet retransmissions, retries to listening to the channel and, energy. Finally, there is a flag of **marked node** that allows the choice of the position of the node in the neighbor table and to identify it as **suspicious route** and locate it as a route at the end of the neighbor table.

To deal with the jamming attacks at the network layer, a mechanism for discerning between the affected nodes and free-jamming nodes should be implemented. Once the affected nodes are identified, the corresponding entries in the routing tables of their neighbors should be handled to avoid the use of routes where such nodes are included. All nodes send HELLO packets for updating the list of neighbors and the modification is based on another information in this packet: performance metrics. If a neighbor node does not answer the HELLO packet then is removed from the routing table. However, the previous strategy could generate too many changes in the routing tables and an excessive overhead. Therefore, the self-configuration processes of the network due to other conditions might produce similar performance degradation in a similar way to the jamming attack, i.e., low-quality links to the neighbor nodes would present a high packet loss rate and collisions. Moreover, to avoid such behaviors where neighbor nodes are being added and removed frequently to and from the routing tables due to causes other than a jamming attack, we studied a proposed mechanism where we include the persistence value p∈{0,1,2,3} associated with each neighbor node. The persistence is increased by one each time that the corresponding neighbor answers the HELLO packet, otherwise is decreased by one. When *p* is equal to zero the node is removed from the neighbor table. We propose three scenarios to try out some variations in the behavior of this mechanism based on the executed routing protocol.

### 3.1. Scenario 1

In this case, the value of persistence parameter grows as fast as decreases. That is, when a node is in a neighbor table of another node acquires the maximum persistence value (p=3). If the node that sends a HELLO packet does not receive an ACK from any of its neighbor nodes, the value of persistence of this node decreases by one, and so on, until drops down to zero and the node is removed from the neighbor table. In this scenario, the node removed from the neighbor table is marked with a confidence flag. Then, if this node responds again to a HELLO packet, enters in the last valid route of the neighbor table but with a persistence of 1 and it may be increasing one by one if its responses to the HELLO packets are satisfactory.

### 3.2. Scenario 2

In this scenario, when a node does not respond to a HELLO packet, it is immediately removed from the neighbor table, i.e., its persistence value goes to zero immediately, not with a soft output. In fact, this scenario is typically implemented by routing protocols, because if a node does not receive an ACK from any route that it has, it makes that route obsolete.

### 3.3. Scenario 3

This scenario has a mitigation technique. There is a *control packet* which is sent by a node based on the review of three parameters: CSMA retries of listening to the channel, packets retransmissions and energy consumption by the node. These three performance metrics are sent as information within the HELLO packet, which is sent when a node wants to know its neighbors. The thresholds of these parameters are calculated and compared with the same parameters of other nodes. This comparison of metrics depends on whether these nodes are directly connected so the metrics are compared with these neighbor nodes, or these parameters are obtained from the perspective of the collector node under an average value of the nodes in the same group of the topology. If a node notes an abnormal performance, while the node compares its performance metrics with those stored in its neighbor table, it can be thought as an “attacked node”. Then, this node itself sends a control packet to its neighbors to get it out of their neighbor tables. Then, this node leaves from the neighbor tables of its neighbors with a flag on “marked node”.

The following algorithm describes the collaborative detection mechanisms. When a node leaves the routing tables there is a reconfiguration topology and a resilience effect of the network according to each routing protocol.

## 4. Analysis of Results

The results are tested in a grid of 7 × 7 with a collector node in the lower left corner, with 5 m between each node and its neighbor horizontally and vertically, see Figure 1. In this figure, we show three shading nodes, representing the three possible points of an attacker node. The traffic rate is 10 packets per second in every node. The jammer node is sending packets under reactive jamming. Table 1 describes the parameters of the simulations.

**Algorithm 1:** Collaborative detection mechanism.**Require** nodes start; **Require** cont = 0;  **for all**
$Node_{i}$
**do**
**Require** set $positionRingNode_{i}$; **Require** Calculate $Energy_{i}$; **Require** Calculate $ReTx_{i}$; **Require** Calculate $CsmaRetries_{i}$; **Require** send HELLO packet with metrics information;   **if** Algorithm == Extended Mechanism **then**
   Collector node sends broadcast to receive information of all nodes;    Collector node sets metric values for each $node_{i}$;    Collector node compares each value per metric for each positionRing;    **while**
$positionRingNode_{i}$ == $positionRingNode_{x}$
**do**
    **if**
Energyi != [Energyxmin,Energyxmax]
**then**
     cont++;     **end if**
    **if**
ReTxi != [ReTxxmin,ReTxxmax]
**then**
     cont++;     **end if**
    **if**
$CsmaRetries_{i}$ != $\left[CsmaRetries_{x}min,CsmaRetries_{x}max\right]$
**then**
     cont++;     **end if**
   **end while**
  **end if**
  **if** Algorithm == Connected Mechanism **then**
   **for all**
$Node_{x}$ neighbor of $Node_{i}$
**do**
    **if**
Energyi != [Energyxmin,Energyxmax]
**then**
     cont++;     **end if**
    **if**
ReTxi != [ReTxxmin,ReTxxmax]
**then**
     cont++;     **end if**
    **if**
$CsmaRetries_{i}$ != $\left[CsmaRetries_{x}min,CsmaRetries_{x}max\right]$
**then**
     cont++;     **end if**
   **end for**
  **end if**
  **IF**
cont>=2
**then**
    $Node_{i}$ sends a control packet to every $Node_{x}$;     $Node_{x}$ deletes $Node_{i}$ from its routing table;   **end if**
 **end for**

WSNs are extensively studied with several network simulators that analyze various performance and energy consumption metrics. However, the use of these simulators aims at the analysis of some topologies or already defined and parametrized environments, as Network Simulator 2 (NS-2) [32]. We can also find network simulators as TOSSIM [33], that estimates the energy consumption while considering the batteries lifetime of the devices and set realistic scenarios with known platforms. This work enables us to build a network simulator based on an event-driven system where we have an approach to the Physical, MAC and, Network layers. Thus, we can simulate any topology, implementing several routing protocols, observe and count collisions, retransmissions, channel retries, and establish models to evaluate energy consumption techniques. Currently available simulators must have the possibility to allow flexibility for modifications to incorporate new protocols. Therefore, we designed and implemented a network simulator based on events using the C++ language. The simulator was conceived with the paradigm of object-oriented programming (OOP), where nodes are autonomous entities (objects) that have properties and functions as transmit, receive, and route packets. Simulation events are managed by a planner which serves as a “task organizer” for objects involved in the simulation. Some advantages of the C++ simulator are the speed of the program execution and the OOP that provides easiness of managing various classes as a separate entities that interact autonomously, for example, a node. We use this event-driven simulator based on C++ language reported in [34]. This is a network simulator with parameters of Physical, MAC and Network layers.

There are two perspectives for forming the groups of nodes that deliver information for the self-aware jamming algorithm: Connected Mechanism and Extended Mechanism. In the Connected Mechanism the nodes are neighbors directly connected, and the values of performance parameters are obtained from them and compared with the actual parameters of the node in evaluation. In the Extended Mechanism the included nodes are all those with similar performance metrics because they are in the same area of the topology or at the same distance from the collector node. In the Extended Mechanism more nodes can be considered, i.e., the whole set of nodes in the same area in the topology with respect to the collector node, which forms a set of nodes with similar performance parameters. The communication to the Extended Mechanism demands higher overhead to the network when the information of performance metrics is gathered, this is because some of the nodes in the groups may not have direct connectivity. In contrast, collecting information within the Connected Mechanism is straightforward. Thus, the latter is a more practical approach, but the former sends the control packet more accurately.

Figure 2 shows the results of the proposed Extended Mechanism for identification of jammed nodes for the MPH protocol. The three possible positions of the jammer node are with respect to the collector node and are: near (position [2,2] of the grid matrix of the Figure 2), middle (position [4,4] in the Figure 2) and far (position [7,7] of the grid matrix of Figure 2). These positions are based on Figure 1. The results show in white squares the nodes that remain in the routing tables of their neighbors and in black squares those that are not included as neighbors, while the gray colors represent intermediate conditions (may be also under attack). This figure has four columns representing four situations in the network: (1) No jamming or normal network conditions, (2) When the jammer node is close to the coordinator, (3) When the jammer node is in the middle of the topology, (4) When the jammer node is in the farthest corner from the coordinator node. The column “No jamming” is represented by the nodes in stable state or normal conditions, therefore, the nodes that are grayer indicate the nodes near to the collector node because they constitute a bottleneck in the network. The nodes that remain white are those that do not have their altered performance parameters. When we obtain the performance parameters in the other cases (near, middle and far) with the jammer node, a difference is established and when the difference is very low, almost zero, the squares are clearer (almost whites) and this indicates that they are remaining under their normal operating conditions. Then, when the difference in the performance parameters (energy, retransmissions, and CSMA retries) is high, in the “near”, “middle” and “far” columns, these nodes become darker. The above represents that in the last three columns, the darker nodes show the affected nodes according to each of the positions of the jammer node. This differential makes the “near”, “middle” and “far” columns are relative, not absolute, values of the evaluation of the three variables on average: energy, retransmissions, and CSMA retries. In this figure the scale is up to 3 because an average estimate of each metric is made and its value is weighted up to 3 so that when we introduce the jammer node, differences are found and hence the colors of the squares come out (which represent the nodes in the network) and we can identify affected nodes. Moreover, an area that is slightly affected in all cases is the proximity to the coordinator node because this becomes a bottleneck because in this type of network all the nodes send packets to the coordinator node, which is located in the lower left corner and, being a square topology, there are few nodes with direct access to this sink node.

Table 2 shows the percentage accuracy of each scenario with respect to the attack zone (where the jammer node should be found). In addition, it presents the percentage of false positives that are nodes incorrectly marked as attacked, but they should not be. This can be due to interference or some problems in the communication channel, but it diverts the attention of the network in the affected zone. The affected area represents the number of nodes that present performance metrics with anomalous values. Remember that the measured parameters for nodes are energy, retransmissions and retries for listening to the channel (CSMA retries). Under any of the two jamming detection mechanisms (Connected or Extended) the nodes compare their metric values, if with any of the considerations of each scheme these values are not similar, this node is marked, and this type of marked nodes make up the affected area. However, in each detection mechanism, these decisions to mark nodes can fail and these nodes that are really affected become false positives. False positives make the affected area confused and the position of the jammer node is harder to find. In the simulation, we know where the jammer node is, but we need each detection mechanism to identify it completely. For this reason, we have chosen 3 positions to locate the jammer node in Figure 1: near to the coordinator node, middle in the topology, and far from the coordinator node. In each of the scenarios we know the number of nodes that surround the jammer node (according to the position it has in the topology), then we can know that for each position the detection mechanism must recognize the affected nodes. Then, the best scenario is the number 3 because the affected nodes are recognized at 100% in the three cases of position of the jammer node and the amount of false positives is much lower than in the other two scenarios (8%, 4%, and 4%). This means that the affected area is delimited with better precision and the affected nodes are clearly recognized.

In Figure 3 we observe the two approaches for gathering the performance metrics information: Connected and Extended mechanisms. These simulations were run assuming Scenario 3, i.e., the scenario with better performance. The results show that we have about 7% better resolution of the affected area under jamming with the Extended Mechanism approach. In this figure, we look for the delimitation of the area where the jammer node is located. There are three positions of the jammer node (near, middle and far) and the darkest nodes are those that are detecting the affected area. The comparisons among performance metrics are done with more information in the Extended Mechanism case, which provides a more assertive decision for sending the control packet. In the Connected approach, the information is only gathered from the directly connected neighbors, so the self-aware algorithm is less efficient. Both mechanisms collect information on node performance metrics that have the same characteristics with respect to proximity to the coordinator node.

We also run a simulation under network stress conditions. With this type of simulation, we want to observe the behavior of both mechanisms in the best simulation scenario (Scenario 3), but in the worst case. One of the cases is a high packets loss in the network due to interference in the links or connections and disconnections of nodes. The other case is due to high traffic of information packets, which increases the overhead along the network and therefore, packet collisions. The previous cases cause that more packets are retransmitted, and the channel is constantly busy, so the CSMA retries also increase. In Figure 4, the first case is represented by packet loss (between 5 and 10% per link) in the links due to possible interference in the channel. Thus, links acquire simulated loss percentages so that we measure if the attack made by the jammer node is camouflaged or not. We observe that the Extended Mechanism has a behavior of around 6% better resolution of the affected area with respect to the Connected Mechanism. In both approaches, it is possible to visualize the area in the presence of the jammer node; however, the Extended algorithm has a better definition.

In Figure 5 network stress is achieved by sending high amount of traffic packets to the collector node (20 packets per second per node). In this figure we also seek to stress the network to observe if the attack of the jammer node is camouflaged due to the increase of values in the performance parameters thanks to the overhead that is being imposed on the network. The difference in the resolution of the area between the two studied mechanisms is 8%, with the Extended Mechanism being of better resolution with respect to the Connected Mechanism.

These two figures show the simulations where the network is under high stress conditions and a possible attack could be camouflaged in this situation. Results show that in the presence of packet loss or high network traffic, considering Scenario 3 which is the most complete because it has detection and mitigation, the Extended Mechanism presents a resolution of the affected area 7% greater than in the Connected Mechanism. Another positive aspect of this analysis is that both proposed mechanisms in this work allow the detection of the affected area, where possibly, the jammer node will be under extreme conditions. The accuracy percentage of affected nodes correctly identified is 75% for the Extended Mechanism and 68% for the Connected Mechanism.

### 4.1. Energy Analysis as a Comparison Metric

Energy is a global metric that influences autonomy in wireless networks. It is important that these networks operate for long periods without human intervention. All performance metrics mentioned in this work impact the energy consumption of the nodes directly or indirectly. Energy consumption is reflected in batteries life and, therefore, in the connections and disconnections of nodes. All these topology changes represent more effort for the routing protocol as the keep the channel busy while the network becomes reconfigured again.

Another contribution of our work is our proposed energy model capable of quantifying the total energy consumed and the energy invested in the main activities of the nodes in a network. This model is based on the time it takes each node to perform the main tasks as a part of a network. This time is reflected into a consumed energy in each activity, and is calculated depending on the parameters involved in each task performed by the node. These times and their voltage and current consumption are listed in Table 3, where the different types of energy considered are shown. Our work considers the energy spent by the MAC and the Network layers. An analytical model based on the CC2530 chip operation [35], system-on-chip (SoC) solution for IEEE 802.15.4, ZigBee, from Texas Instruments is proposed here. This model describes the energy used for each of the functions performed by a CC2530 chip, from the moment that a node is on and is part of the network, as it receives and sends messages, runs MAC layer algorithms, and changes of state, to the time the node is off.

This model includes the main types of energy of the basic activities that sensors carry out in a common performance in a network. These types of energy are: Microcontroller energy $\left(E_{MC}\right)$, starting-up energy $\left(E_{Starting}\right)$, shutdown energy $\left(E_{Shutdown}\right)$, CSMA/CA algorithm energy $\left(E_{CSMA}\right)$, switching energy $\left(E_{Switching}\right)$, transmission energy $\left(E_{TX}\right)$ and receiving energy $\left(E_{RX}\right)$. All these energies are expressed in Joules.

Therefore, the total energy can be calculated with Equation (Equation 1)
(1)$$\begin{array}{cc} E_{Total}= & E_{MC}+E_{Starting}+E_{Shutdown}\\ & +E_{CSMA}+E_{Switching}+E_{TX}+E_{RX}. \end{array}$$

The energy model described by the above equation measures the energy used by the main functions of a node belonging to a sensor network, assuming that the functions related to the wireless communications are the most relevant form an energy point of view.

We take into account that one of the main effects of attacks on WSNs is the increase in consumption in the attacked nodes, which reduces the useful life of the node and the network. Therefore, it is essential to estimate the consumption impact of both the attacks and the strategies adopted to improve the system security. Table 4 shows the energy consumption for each of the node’s main activities at network level. We make this comparison to base the consumption on the energy model proposed for the sensors, detailed in Table 3. Each of the energies described here are total energies for 30 s of simulation time and based on the grid described in Figure 1. We expose the two techniques of jamming detection when the jammer node is in the middle of the topology, jamming is reactive, and the percentage of packet loss per link is between 0.5 and 2.5%. We note that the Extended Mechanism improves energy consumption by about 20% against Connected Mechanism. This table checks the energy consumption of the nodes under the reactive jamming, which is the most aggressive of all the cases for the scenarios we have proposed. We analyze each one of the energy types of the model to visualize where the greatest energy expenditure is made at the moment of a network under attack conditions. The types of energy that have the greatest impact are the energy of the CSMA algorithm, the receiving energy and the transmission energy. This situation is logical because, under attack, the amount of overhead in the network increases and, therefore, increase packet retransmissions, retries to listen to the communication channel and collisions.

### 4.2. Extrapolation of the Proposed Mechanism for Irregular Traffic

In this case, we analyze a bigger topology network with 225 nodes where these have different traffic generation functions. Traffic of each node is obtained from a given interval between low and medium traffic function (1–10 packets per second).

Figure 6 presents the shaded area for the three performance metrics the Scenario 3. Thanks to the simulation results obtained so far, it is important to scale the model to have a reproduction of a medium network and be able to analyze the behavior of each of the performance metrics: retransmissions, CSMA retries, and energy. After careful observation and analysis of a matrix of 225 nodes we noticed that the method of detection and mitigation used can be extrapolated to a greater number of nodes. We expose the Extended technique for jamming detection when the jammer node is in the middle of the topology, jamming is reactive, and the percentage of packet loss per link is between 0.5 and 2.5%. The simulations are run for three characteristic protocols that represent different approaches to know routes in the network: reactive protocols (AODV and DSR), proactive protocol (MPH). In this way, we have a broader analysis of the Extended Mechanism’s impact on different communication protocols and a greater number of nodes. The darker area shows the possibility of finding the jammer node there. Please note that in the proactive protocol, such as MPH, the delimitation of that area is reduced to 10% of the network, whereas in reactive protocols, such as AODV and DSR, it is reduced to 25% of the complete topology.

### 4.3. Comparison among AODV, DSR, and MPH Routing Protocols

As discussed above, energy is an important metric for studying the distribution of resources at network nodes. Figure 7 shows the energy per node when the network is under normal conditions and when the network is under attack and the attacking node is in the middle of the topology (Position 24 in Figure 1). Remember that we have 49 nodes in a square topology, and we want to know the total energy (in Joules) per node. When the network is under normal (stable) conditions, nodes 1, 7 and 8 present a normal increase in energy consumption because they are the closest nodes to the coordinator node and, because of the traffic distribution packets of the rest of the nodes in the network. Now, when the network is under jamming, the nodes that have the greatest impact on energy consumption should be 16, 17, 18, 23, 25, 30, 31, and 32, because it is the area of those directly connected with the jammer node. Indeed, this happens for the three studied protocols and the peaks that are observed in the graph just the levels corresponding to these mentioned nodes, including the jammer node, because this node is running under a reactive jamming where the energy consumption is higher. As shown, the total average energy for AODV is 0.1209, for DSR is 0.1105 and for MPH is 0.0397. Therefore, the MPH protocol is more energy efficient by 74% than AODV, and 69% than DSR considering the average energy of the nodes.

In Figure 8, we compare the performance in terms of the persistence parameter of MPH and two others widely known protocols, AODV and DSR. Remember that the persistence parameter, *p*, indicates the soft exit of the nodes from the neighbor tables of the other nodes, depending on the scenario (previously we mentioned Scenario1, 2, and 3). This exit of nodes from the tables is due to the distrust that this node has because its performance metrics have anomalous values. Another point worth noting is that the Figure 8 describes the location of the jammer node for three different positions (near, middle, and far from the collector node). Simulations were performed considering Scenario 3 and modifying AODV and DSR protocols by including the transmission of a control packet. This control packet is sent in Scenario 3 to inform a node that it must remove another from its neighbor table, to no longer have that route as a path in the network. The nodes deleted through this control packet seek to isolate the possible jammer node without it notices and, for this reason, will continue to carry out its attack. The disadvantage that could arise in this scheme is some network routes are lost, because we disconnect some nodes (the nodes directly connected to the possible jammer node); however, the mitigation of the attack will not impact on the other nodes and the energy of the jammer node can be consumed more quickly. The results show that AODV and DSR can reduce the area of the location of the jammer node to approximately 15% of all the network while MPH can further reduce the area to 4%. This ensures that the attacker node may be better located under a possible jamming with a reduced zone. The smaller the area, the more precise the location of the jammer node and the fewer nodes in the network will be disconnected.

Table 5 complements the analysis of Figure 8 under the Scenario 3. These performance metrics are well known in the WSN. This Table has a simulation time of 100 s and analyzes the reactive jamming of a jammer node located in the middle of the topology. The Table analyzes different performance metrics such as: retransmissions, CSMA retries, end-to-end delay, resilience, affected nodes, and energy. Retransmissions are the times that packets that are not received, and they are sent again. We observe that the MPH protocol has a 28% lower retransmissions with respect to AODV and DSR. For CSMA retries, MPH has 18% fewer attempts to listen to the channel than AODV and DSR, under a state of a jamming reactive. Regarding end-to-end delay, packets under the MPH protocol arrive at the coordinator node 0.51 s faster than packets in AODV and DSR. These three metrics indicate that the MPH protocol has better reactivity to isolate the attack of a jammer node against AODV and DSR. Valid routes of MPH are faster than those of AODV and DSR. However, AODV is still that presents the metrics with the lowest performance. The resilience is the ability of the network to recover from an abnormal state, in this case we run the simulation for the same 100 s and the jammer node is activated in second 30, for 10 more seconds, and then disappears. We want to observe how the network is recovered under each of the protocols and we note that MPH has better resilience in 18% with respect to AODV and DSR. The metric of affected nodes is measured in nodes that change their values of metrics: energy, retransmissions, and CSMA retries, in an anomalous way. This metric is closely related to false positives, because they are collateral effects that are generated according to each protocol under jamming. Finally, the MPH protocol has a lower energy consumption of 58% with respect to AODV and DSR. This represents that MPH has a very good recovery reactivity when the network is under attack, specifically, an attack in a focused area at almost the same distance from all nodes in the network (including the coordinator node). In other words, because of the hierarchical and proactive nature of MPH protocol, it presents a rapid recovery from a reconfiguration of the topology, the lost packets are less, and the delay is smaller compared with AODV and DSR protocols. Moreover, the delimitation of the affected area presents fewer nodes, i.e., it is more accurate for finding the jammer node in it.

The advantage that Scenario 3 presents, which is the best scenario chosen and tested in most simulations, is that routes of possibly affected nodes are not eliminated immediately, but that the nodes present a smooth exit from the routing tables of other nodes. These consequences of route isolation are analyzed when we study the resilience in each of the protocols. The recovery capacity of the network shows how you can get to re-establish routes that had been isolated and how it impacts the fact of isolating them in the performance of the network. Then, the proposed detection mechanisms (Connected and Extended) do not completely eliminate the affected nodes (or what is the same, the affected routes), but it locates these routes as less secure. The above is clearly observed in that none of the protocols completely loses their route redundancy (for example, we have a small network of 49 nodes) and the network resilience capacity in none of the protocols decreases by 75%.

Figure 9 shows a direct comparison between the three protocols under the two proposed jamming techniques, based on the total energy for each of the three protocols. This figure is very interesting because it shows the energy performance of each node in each of the three protocols studied. The simulation was made for 100 s, under Scenario 3 and the jammer node is in the middle of the topology emitting a reactive jamming. Again, we note that nodes 1, 7 and 8 are the nodes that forward all network traffic to the coordinator node, and because of the topology, these nodes have a high traffic load and their performance metric values increase more than in the other nodes in the network. Nodes 16, 17, 18, 23, 25, 30, 31, and 32 are those that directly limit the jammer node. These nodes present different behavior than the rest of the nodes of the network and, therefore, their energy increases in a noticeable way. First of all, it is easy to note how MPH has a better performance than DSR and AODV with a value of 0.0390 on the Extended Mechanism and 0.0306 on the Connected Mechanism since DSR has a value of 0.0758 and 0.1062 and AODV has a value of 0.0984 and 0.1147 on the Extended and Connected mechanisms respectively.

In Figure 10 we represent the energy values for each of the three protocols in a box and whiskers scheme. A graph of this type consists of a rectangular box, where the longer sides show the interquartile path. A vertical segment that indicates where the median is positioned and therefore its relation to the first and third quartiles divides this rectangle. This box is located on a scale on a segment that has as ends the minimum and maximum values of the variable. The lines that protrude from the box are called whiskers. These whiskers have a prolongation limit, so any data or case that is not within this range is marked and identified individually. On the bottom of the box is shorter than the one on the right in the three studied protocols; so, 25% of the nodes that spend less energy are more concentrated than 25% of those which consume more. In addition, for the three protocols, the energy consumed by nodes between 25% and 50% of the population is more dispersed than between 50% and 75%. We observe that the displacement of the box plots downwards, especially in the MPH protocol, indicating that here most nodes spend less energy in both mechanisms, being the difference between the maximum and the lowest minimum, as well as the interquartile difference. The results show that in the AODV protocol the energy distribution is more dispersed than in the other two protocols and there is a higher energy expenditure compared to the other two protocols. For the MPH protocol, there is more atypical data than in the other two protocols because in both mechanisms, the nodes that surround the jammer node consume more energy, a parameter that is very noticeable to isolate an area of the network.

Table 6 shows values of performance metrics along the network. This table describes a comparison of the three protocols studied with the modifications made in each detection mechanism jamming (Extended and Connected). The tests were made with non-ideal links and traffic rate of 10 packets per second per node under the grid topology shown in Figure 1. Metrics are average values of retransmissions and CSMA retries. The delay end-to-end is calculated as the average per route. Overhead is a metric that influences the amount of network collisions and the channel occupancy. Overhead indicates the number of control packets under a routing protocol. It depends on the number of routing packets that the routing protocol needs to connect nodes and to route packet traffic. Thus, it is calculated taking into account the number of control packets that are needed to route a traffic packet. Ideally, a routing protocol should need the least amount of control packets. A metric such as energy is given by the average energy per node along the network. Resilience is a parameter that measures the recovery time of a system. This metric is analyzed under the network under normal conditions and after 30 s the jammer node is introduced in the middle of the topology and after 10 s the jammer node goes out and here we observe and measure how long it takes the network to recover and return to its normal conditions. The affected nodes are a metric that is set as a ratio of nodes that appear altered in their performance parameters and are those that are actually encircling the jammer node between nodes that ideally should surround the jammer node completely. For this grid theoretically establish 9 nodes that should completely surround if the jammer node is in position 24 of the topology in Figure 1.

### 4.4. Performance in a Different Network Structure

To give stringency to the proposed scheme, simulations should be extensive in terms of analysis of the performance in different network structures. Because of the scheme that has the best results is Scenario 3, we take into account 4 protocols with different characteristics in WSNs: AODV, DSR, MPH, and ZTR. Please note that AODV and DSR are reactive protocols, MPH is a hybrid protocol between proactive and reactive, and ZTR (Zigbee Tree Routing) [37] is a proactive hierarchical protocol. ZTR is a simple protocol which establishes parent-child links and the nodes always carry information to their parent. It has a tree topology and is easy to implement. ZigBee requires that there is at least one full-function device with a more robust nature to act as a network coordinator, but the final nodes can have reduced function to reduce costs. The parent node is the one which has given the child access to the network, so parent-child links are created, but each child can only have one parent. Some of the advantages of ZTR are that in the algorithm implemented in the network layer, there is a balance between cost per unit, battery expense, complexity of implementation to achieve a proper cost-performance relation to the application [38].

Regarding the structure of nodes, in the non-uniform random distribution, the coordinator node is in the center of the topology, and the remaining nodes are distributed along the area. We have 100 nodes in evaluation (with the collector node). Sixty percent of the nodes, including the jammer node, are located near the central part of the topology, and the rest of the nodes, 40%, are distributed in the remaining area. The area is 50 m × 50 m.

Regarding the retransmissions, MPH has 16% fewer retransmissions per node than AODV, 13% less than DSR, and 3% less than ZTR. Regarding the CSMA retries, MPH exhibits an average reduction of 13% against AODV, 11% against DSR, and 2% than ZTR. The good news is that these two metrics are broadly related to network redundancy, and MPH performs well, much like an algorithm as simple and fast as ZTR, which is good. MPH also has hierarchical characteristics and adds the multi-parent concept, which generates more redundancy and fewer retransmissions and CSMA retries. Concerning the overhead, MPH and ZTR protocols have a similar behavior, due to their hierarchical structure, while reactive protocols have more routes and more types of packets. It is logical that the ZTR protocol has the lowest energy expenditure of the four protocols, because it is the simplest. MPH only has 2% more energy consumption than ZTR. It is important to mention that Extended Mechanism continues presenting approximately 3% improvement in overall performance in the 4 studied protocols.

### 4.5. Comparison with Another Jamming Detection Algorithm

To compare our proposal with the algorithm cited in [19], we will explain better the mechanism mentioned in this work as a jamming detection method. When any of the mobile nodes or base-stations detect an attack, the node changes its frequency based on a pseudorandom sequence. If no attack is detected, it checks if the node is a base station or if it is a real node. One of the similar characteristics with our algorithm is that in both the associated nodes are informed about changing their state. In the algorithm quoted in [19] the reported nodes change their frequency of operation. In the mechanism proposed in this paper (connected and extended algorithms), the informed nodes change their parameters to make a detour of the affected area. Continuing with the algorithm cited in [19], in the case that the node is a base station when a node has not been informed of the attack, beacons are sent to the associated nodes, and the nodes that have not responded are searched. In the case where the node is an authentic node, if it received the change of frequency by the access point, its frequency changes. Otherwise, it listens for beacons. Then, if the beacons are not received, the node changes its frequency of operation, otherwise, it responds to the beacon. Table 7 describes the main features of each algorithm mentioned before.

We have a scenario that consists of 5 nodes located in a star topology and a coordinator (6 nodes in total) to test the comparison of algorithms. This arrangement is described in Figure 11. To create a ZigBee network, it is necessary to use the CC2531 as a coordinator because the CC2650’s flash is not enough to build the network. The CC2650 LaunchPad includes: External 8Mbit Serial Flash for supporting Over the Air (OAD) firmware updates, 2x push buttons, complete contents of CC2650EM-7ID, the evaluation module, 2x LEDs, 2x BoosterPack connectors, standardized LaunchPad form factor XDS110 debugger with external target interface. We use the 802.15.4 protocol. Easy link is a protocol for transmitting packets and creating networks through the 2.4 Ghz frequencies. It is also possible to use the CC2650 as a sniffer, to acquire the ZigBee packages. In addition, a CC2530 node from Texas Instruments has been used as a coordinating device. The evaluation board used to program it is the SmartRF05 evaluation board.

The system parameters for the experimentation setup are:Experimentation area ($m^{2}$): 50.Transmission range (m): 15–18.Dimension between nodes (m): 10.Packet rate (kbps): 250.Protocol: ZTR.Packet size (bytes): 100.Number of coordinators: 1.Number of attackers: 1.Number of available channels: 11.Experimentation time (sec): 100.

Table 8 shows a great similarity between performance metrics, taking into account that originally our proposed algorithm aimed at a scenario of static nodes. We compared some performance metrics with a literature algorithm to analyze the benefits of our best evaluated scheme in this study: the Extended Mechanism. However, the Extended Mechanism proposal shows very good performance in terms of overhead (approximately 17% less overhead than the other two proposals) and the jamming duration (approximately 18% less duration than the other two proposals). Likewise, the proposal of the Extended Mechanism has a performance lower than the algorithm cited in [19] of approximately 6%. Regarding energy consumption metric, we measure in both cases the total network consumption during the experimentation time. The best scenario that our work presents is the Extended Mechanism, which has a consumption of 7% less energy with respect to the algorithm evaluated in [19].

## 5. Conclusions

In recent years, a widespread requirement of WSNs is the assessment of security, especially because attackers can corrupt the network, accessing or modifying information and affecting the behavior of nodes. Due to the low-cost and low-power requirements of these types of systems, nodes usually have limitations in both hardware/software and computing capacities. This is why the design of security in WSNs must take into account the memory, computational capacity and availability of hardware/software resources. In addition to this, energy consumption is limited. Usually, power consumption is one of the most significant constraints that designers have when developing WSNs.

In our work, we propose a detection mechanism that marks the affected area and does not allow the attack to propagate. Our mechanism is based on cooperation, namely by receiving feedback from neighbor nodes. The mechanism was tested for reactive jamming, which is considered a worst-case scenario in jamming attacks. In addition, the methodology was evaluated under two extreme situations, namely by considering high traffic and high packet loss due to link interference. Thus, we described and compared two different jamming detection techniques named Connected and Extended mechanisms. Both take advantage of the intrinsic characteristics of the WSN which mainly reside on the routing protocols that are designed to be fast and efficient. However, the Extended Mechanism exhibits a better performance (nearly 20%) than the Connected Mechanism, the main reason behind it is that the Extended one requires a collector node to be robust and have good connectivity to the network, in other words, it has to be a root node. Therefore, the Connected Mechanism is a better choice due to the easiness to use and implement it, even though it has fewer parameters for self-assessment.

Results show that the proposed mechanism is more efficient in a hierarchical protocol such as MPH. Moreover, the Extended approach presents better accuracy compared to the Connected Mechanism approach due to its global character and the fact that the collector node has the perspective of all nodes with similar performance features. Finally, the scalability of the proposed mechanism was tested in a 225-node grid, showing that the affected area (i.e., the accuracy) can be reduced to 10% of the total area with an MPH proactive protocol and 25% with AODV and DSR protocols.

## Figures and Tables

**Figure 1 sensors-19-02489-f001:**
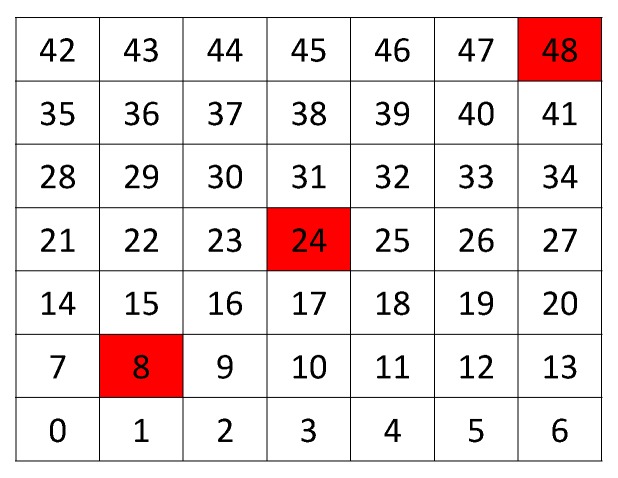
Network grid.

**Figure 2 sensors-19-02489-f002:**
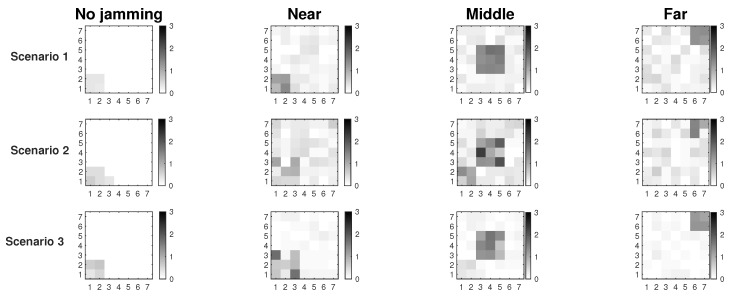
Three scenarios using the detection technique over MPH protocol.

**Figure 3 sensors-19-02489-f003:**
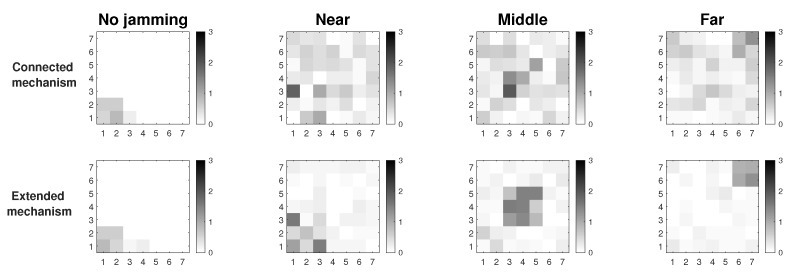
Connected vs extended mechanisms over MPH protocol under the Scenario 3.

**Figure 4 sensors-19-02489-f004:**
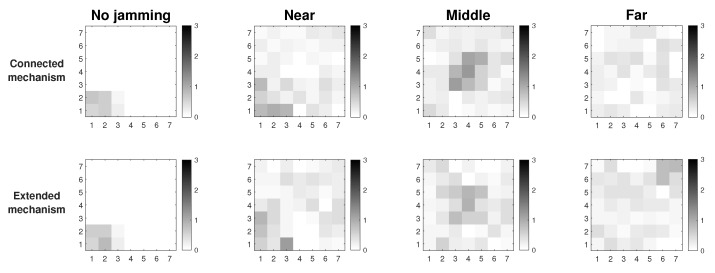
Connected vs extended mode under maximum packet loss.

**Figure 5 sensors-19-02489-f005:**
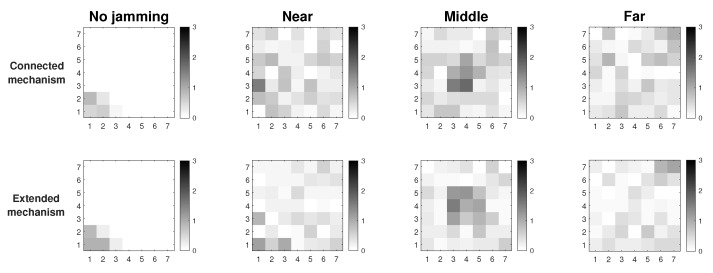
Connected vs extended mode under high traffic.

**Figure 6 sensors-19-02489-f006:**
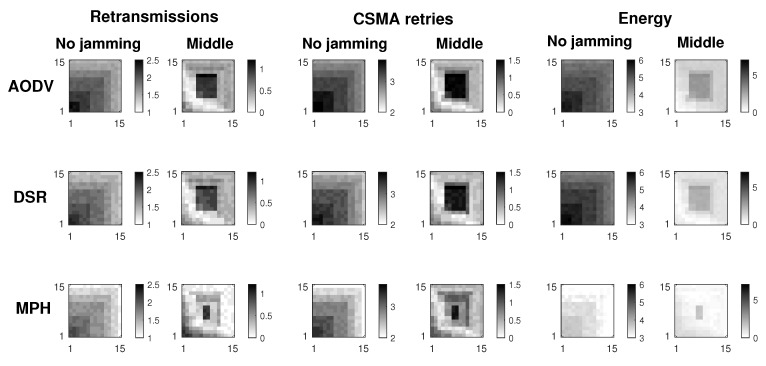
Network links with irregular traffic.

**Figure 7 sensors-19-02489-f007:**
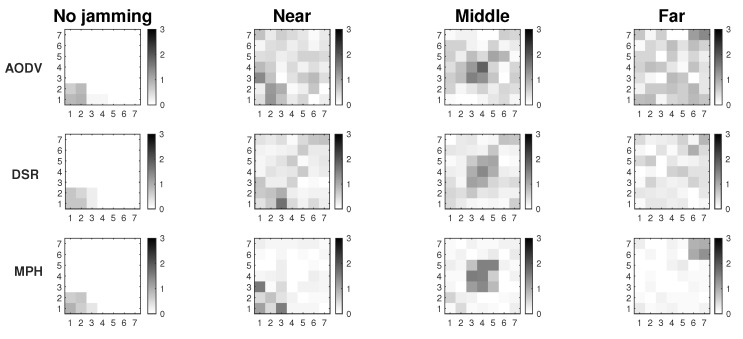
Comparison of energy per node under stable and attacked conditions.

**Figure 8 sensors-19-02489-f008:**
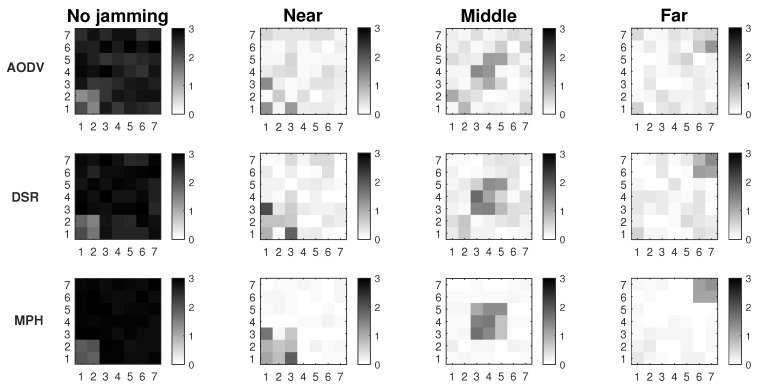
Comparison for AODV, DSR, and MPH protocols under Scenario 3.

**Figure 9 sensors-19-02489-f009:**
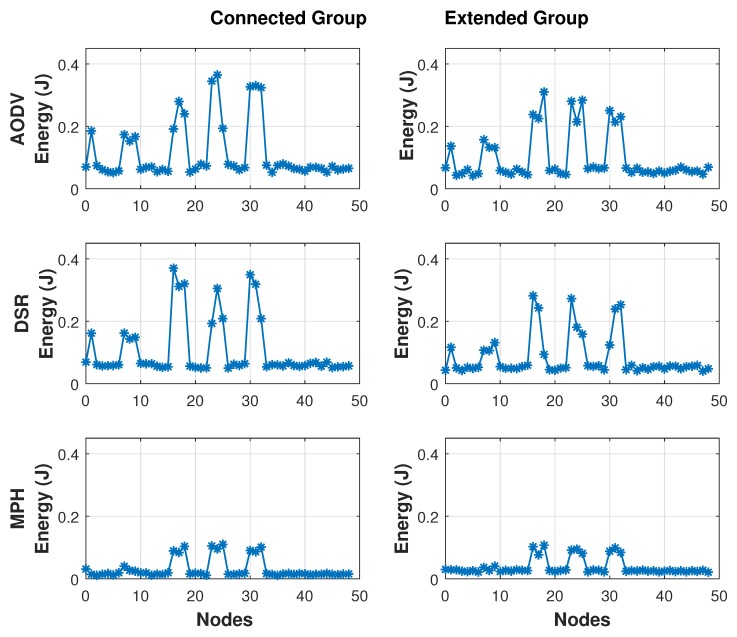
Energy per node under each protocol.

**Figure 10 sensors-19-02489-f010:**
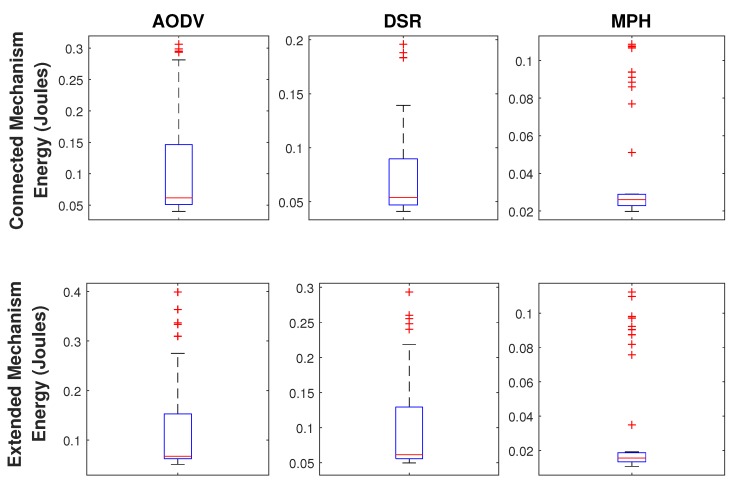
Comparison of energy distribution for AODV, DSR, and MPH protocols under Scenario 3.

**Figure 11 sensors-19-02489-f011:**
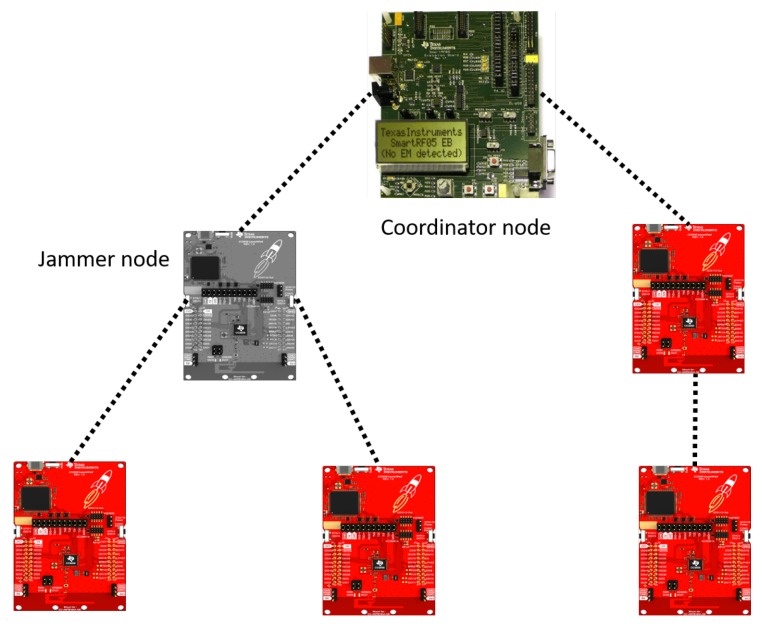
Network configuration for comparison of algorithms under experimentation.

**Table 1 sensors-19-02489-t001:** Simulation and real network parameters. CSMA/CA, carrier sense multiple access with collision avoidance [31].

Parameter	Value
**Physical Layer Parameters**	
Sensitivity threshold receiver	−94 dBm
Transmission power	4.5 dBm
Propagation model	Free Space
**MAC Layer Parameters**	
Waiting time for ACK packet	30 ms
Maximum retransmission number	3
Maximum retry number	5
Maximum number of tries to reach a node from the collector	9
Packet error rate	1%
Average frame length	22 bytes
Maximum number of backoffs	4
MAC protocol	IEEE 802.15.4
MAC layer	CSMA/CA
**Network Layer Parameters**	
Number of nodes	49
Maximum number of neighbors	16
Discovery neighbor time	30 s
Update time neighbors table	30 ms
Maximum data rate	250 kbps
Routing	Hierarchical
Scenario	Static nodes

**Table 2 sensors-19-02489-t002:** Detection of affected zone and false positives.

Position	Scenario 1		Scenario 2		Scenario 3	
	Affected Area	False Positives	Affected Area	False Positives	Affected Area	False Positives
Near	100%	18%	75%	29%	100%	8%
Middle	100%	8%	100%	22%	100%	4%
Far	100%	8%	75%	39%	100%	4%

**Table 3 sensors-19-02489-t003:** Energy Model [36].

	Voltage (mV)	Current (mA)	Time (ms)	Energy (J)
Start-up mode	120	12	0.2	$E_{Starting}=0.000288$
MCU (32-MHz clock)	75	7.5	1.7	$E_{MC}=0.000956$
CSMA/CA algorithm	270	27	1.068	$E_{CSMA}=0.00778$
Switch from RX to TX	140	14	0.2	$E_{Switching}=0.000392$
Switch from TX to RX	250	25	0.2	$E_{Switching}=0.00125$
Radio in RX mode	250	25	4.1915	$E_{RX}=0.0262$
Radio in TX mode	320	32	0.58	$E_{TX}=0.00426$
Shut down mode	75	7.5	2.5	$E_{Shutdown}=0.00141$

**Table 4 sensors-19-02489-t004:** Energy consumption for each detection mechanism.

Type of Energy	Connected Mechanism Energy (J)	Extended Mechanism Energy (J)
$E_{Starting}$	0.0172	0.0144
$E_{MC}$	0.1338	0.0960
$E_{CSMA}$	0.8021	0.7982
$E_{Switching}$	0.0509	0.0411
$E_{RX}$	0.2156	0.2268
$E_{TX}$	0.1062	0.0923
$E_{Shutdown}$	0.0809	0.0749

**Table 5 sensors-19-02489-t005:** Performance metrics under attack and Scenario 3 when the jammer is in the middle of the topology.

	Retransmissions	CSMA Retries	Delay End-to-End (sec)	Resilience Capacity (%)	Affected Nodes	Energy (J)
**AODV**	2.52	3.44	1.398463	80	18	6.4
**DSR**	2.47	3.33	1.284773	75	13	6.0
**MPH**	1.93	2.60	0.811836	85	9	3.9

**Table 6 sensors-19-02489-t006:** Performance metrics under Extended Mechanism and Connected Mechanism for a non-uniform random distribution.

	Retransmissions	CSMA Retries	Delay End-to-End (sec)	Resilience (sec)	Overhead (%)	Energy (J)
**AODV-M**	2.92	4.44	1.473984	8.1	50	6.82
**AODV-under Connected Mechanism**	2.30	4.12	1.258736	7.3	46	6.30
**AODV-under Extended Mechanism**	2.13	4.02	1.190286	7.1	43	6.22
**DSR-M**	2.80	4.30	1.387409	7.3	43	6.51
**DSR-under Connected Mechanism**	2.27	3.98	1.248273	5.0	39	6.23
**DSR-under Extended Mechanism**	2.16	3.90	1.188223	4.8	35	6.17
**MPH-M**	2.44	3.82	0.687308	3.2	28	4.08
**MPH-under Connected Mechanism**	2.14	3.60	0.518376	3.0	25	3.86
**MPH-under Extended Mechanism**	2.06	3.49	0.441029	2.8	22	3.70
**ZTR-M**	2.50	3.90	0.598461	2.8	30	3.89
**ZTR-under Connected Mechanism**	2.55	3.88	0.568326	2.5	26	3.72
**ZTR-under Extended Mechanism**	2.52	3.83	0.548705	2.6	25	3.68

AODV-M is modified AODV; DSR-M is modified DSR; MPH-M is modified MPH; ZTR-M is modified ZTR.

**Table 7 sensors-19-02489-t007:** Contributions of each algorithm.

Feature	Proposed Mechanism in This Work	Algorithm Cited in [19]
New type of nodes	Marked nodes	Honeynodes
Function of nodes	Encircle the jammer node	Try to attract jammers
Protection of the coordinator node	Isolate the attack until the jammer node decreases its energy	Hybrid proactive and reactive frequency selection
Operation	Proactive and reactive jamming	Proactive and reactive jamming
Detected attack	The nodes affected by their performance metrics isolate the jammer node	Predefined pseudorandom sequence for the selection of the next frequency

**Table 8 sensors-19-02489-t008:** Comparison of the algorithms: presented in this work and the cited in [19].

Algorithm	Jammed Duration (sec)	Message Overhead(%)	PDR(%)	Energy (J)
Connected algorithm	2.2	100	94.0	2.345
Extended algorithm	1.7	80	95.1	2.027
Algorithm cited in [19]	2.1	94	96.3	2.158

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
