# Peer review of "New Detection Paradigms to Improve Wireless Sensor Network Performance under Jamming Attacks"

_sensors, 2019, doi:10.3390/s19112489_

Reviewer 1 Report

Paper summary:

The authors propose a detection mechanism for reactive jamming for sensor networks. The aim of the propositions is to isolate the affected zone from the safe one. The authors propose two mechanisms, where the first one uses direct nodes to cooperate and send the evaluation parameters, named connected paradigm. While the second mechanism, named extended mechanism, aims to use all the nodes evaluations and send them to a collector node. The parameters used for detecting the jamming interference are retransmissions, channel activity,  and energy . The detection is based on comparison of the parameters between the neighbors nodes for the first mechanism and nodes belonging to the same group for the second mechanism.

In order to detect the jamming nodes, the authors introduce  a parameter named persistence parameter to be added in the neighboring table of routing protocol in order to avoid the affected nodes.

The authors evaluate the proposed mechanisms using different routing protocols. Namely, MPH which is previously proposed by the authors, in addition to AODV and DSR. In the last part of the simulations, they also use ZTR protocol.

Weak points:

-The authors do not compare their work with other jamming detection works but rather show the advantage of their already published work MPH compared to AODV and DSR. It is required to choose a same routing protocol for different detection methods and compare them together. Or compare the proposition with a routing work that handles jamming. The authors present in the related work section many works that could be considered as candidates in the simulation comparison.

-The paper is not well written and needs to be proofread. The current version contains a lot of typos.

-The paper needs also to be reorganised. For example in the related work section, the authors explain some parts of their proposal, and come back again to other related work. It should be done separately, unless using the same behaviour of a part in a described work.

-Why the authors use their own simulator rather than the known ones like tossim and contiki, it is better to motivate this choice in few words.

-The tables and Figures are not well referenced, in many occurrences there is ??.

-The algorithm 1 is not well written.

-The table 2 should be explained better as the explanation is not sufficient. The same remark concerns all the figures and tables results.

-Why the scenario with no jamming in figure 3 is black (and also other figures). It should not be white for the cause of absence of jammers? Perhaps I don’t understand well the mechanism, for that I propose more explanations. In general, for the different figures I don’t understand the relation between the no jamming scenario and the black boxes . I think that without jamming, it should be white, not?

-The caption of figure 7 does not match with the explanation of section 4.3

-Generally, all the figures need more analysis to explain the results, as it is not well done.

-The bibliographic references need also to be more organised. Some are duplicated like 12 and 16.

-Do the authors evaluate the consequences of isolating the affected routes in case there is no other routes, rather than choosing the best of the existing ones (the less affected)

Here some typos:Please proof read the paper

-introduction:

   * Among then ==>  Among them

   *AODV uses a routing table to known destinations to known routes. ==> change

  * it is included in said message to keep track ==> in the said message

  *we introduce a hybrid protocol (between proactive and reactive nature), named MPH (Multi-Parent Hierarchical) [14]. It was designed and implemented in the reference cited in [14] … ==> change

* reactive protocols [11] are heavily used in WSNs since they create a small overhead of packets on the network and only create nodes when deemed necessary ==> create nodes, or nodes entries in routing tables?

* AODV uses a routing table to known destinations to known routes. ==> this sentence should be reformulated.

* “...where the hierarchy of the nodes is given by its location level...” ==> …by their location..

related work section:

* “such packet is used for informing to the neighbor nodes that there is a jamming attack in progress” ==> ...informing the neighbor nodes...

 Analysis of detection mechanism section:

 *”the modification is based on there is another information in this packet...”==> ...is based on another information…

 *” This three performance metrics are sent...” ==> these three…

 *” If a node notes an abnormal performance, that it is the node compares its performance...”==> …. performance while the node compares...

 Analysis of results, Energy analysis as a comparison metric section

 *” Energy is a global metric that influence autonomy...” ==> ...influences...

*” This energy consumption is reflected in the lifeof the batteries lifeand,...” ==> change

*” All of these topology changes represent more effort for the routing protocol as the keep the channel busy while the network becomes reconfigured again.” ==> change

*” These times and its voltage and current consumption are listed ...”==> … and their voltage...

*”Each of the energies described here are total energies for 30 seconds of simulation time and based on the grid described in 1 “ ==> … described in Figure 1

*” At first, is easy to note how...”==>  At first, it is easy ...

Bibliography references should be proof read. For exemple:

  [14]: Del-Valle Soto, C.; Mex Perera, C.; Orozco Lugo, A.; Galvan Tejada, G.M.; Olmedo, O.; Lara, M. Anefficient Multi-Parent Hierarchical routing protocol for WSNs. Wireless Telecommunications Symposium,2014.Cited By :5.==>to delete :Cited By :5.

[20]Lou, L.; Fan, J. An anti-jamming routing selection criteria based on the cross-layer constraints of channelstate information for MANETs. Advanced Information Technology, Electronic and Automation ControlConference (IAEAC), 2015 IEEEIEEE, 2015, pp. 1000–1004.

Author Response

Dear Editor,

Dear Sensors Editorial Office,

We are submitting the paper:

“New Detection Paradigms to Improve Wireless Sensor Network Performance under Jamming Attacks”

Authored by: Carolina Del-Valle-Soto *, Carlos Mex-Perera , Ivan Aldaya , Fernando Lezama , Juan Arturo Nolazco-Flores , Raul Monroy

We would like to thank the reviewers and editors for their detailed analysis of the manuscript; the comments are very valuable to us. In the revised version of the paper we have incorporated the changes recommended by the reviewers.

Comments to all observations and suggestions including point-by-point responses are addressed in the following text.

_______________________________________________________________________________

Reviewer 1 comments

Comment 1: The authors do not compare their work with other jamming detection works but rather show the advantage of their already published work MPH compared to AODV and DSR. It is required to choose a same routing protocol for different detection methods and compare them together. Or compare the proposition with a routing work that handles jamming. The authors present in the related work section many works that could be considered as candidates in the simulation comparison.

Response: We thank the reviewer for his/her observations and comments that have contributed to the improvement of the paper.

We added the following paragraph in the Related Work section in order to introduce the algorithm to compare our detection mechanisms in other section:

Jamming is considered one of the most harmful attacks in threats to WSNs. A jamming attack can limit the communication capabilities of a WSN by degrading the system performance interfering with signals using certain interfering devices [18]. For instance, the proposal in [19] is based on a jamming detection algorithm that employs honeynodes and response mechanism concepts. Each node cooperates with the transmitter to detect any jamming. The authors analyze network performance metrics such as: duration of jamming attack, control message overhead, and the number of channel re-configurations, as well as One of the metrics they use in their simulations is the Packet Delivery Ratio (PDR).

Moreover, we added a subsection called “Comparison with Another Jamming Detection Algorithm”, as follow:

In order to compare our proposal with the algorithm cited in [19], we will explain better the mechanism mentioned in this work as a jamming detection method. When any of the mobile nodes or base-stations detect an attack, the node changes its frequency based on a pseudorandom sequence. If no attack is detected, it checks if the node is a base station or if it is a real node. One of the similar characteristics with our algorithm is that in both the associated nodes are informed about changing their state. In the algorithm quoted in [19] the reported nodes change their frequency of operation. In the mechanism proposed in this paper (connected and extended algorithms), the informed nodes change their parameters to make a detour of the affected area. Continuing with the algorithm cited in [19], in the case that the node is a base station when a node has not been informed of the attack, beacons are sent to the associated nodes, and the nodes that have not responded are searched. In the case where the node is an authentic node, if it received the change of frequency by the access point, its frequency changes. Otherwise, it listens for beacons. Then, if the beacons are not received, the node changes its frequency of operation, otherwise, it responds to the beacon.

Table 8. Contributions of each algorithm

We have a scenario that consists of 5 nodes located in a star topology and a coordinator (6 nodes in total) in order to test the comparison of algorithms. This arrangement is described in Figure 11. In order to create a ZigBee network it is necessary to use the CC2531 as a coordinator because the CC2650's flash is not enough to build the network. The CC2650 LaunchPad includes: External 8Mbit Serial Flash for supporting Over the Air (OAD) firmware updates, 2x push buttons, complete contents of CC2650EM-7ID, the evaluation module, 2x LEDs, 2x BoosterPack connectors, standardized LaunchPad form factor XDS110 debugger with external target interface. We use the 802.15.4 protocol. Easy link is a protocol for transmitting packets and creating networks through the 2.4Ghz frequencies. It is also possible to use the CC2650 as a sniffer, to acquire the ZigBee packages. In addition, a CC2530 node from Texas Instruments has been used as a coordinating device. The evaluation board used to program it is the SmartRF05 evaluation board.

Figure 11. Network configuration for comparison of algorithms.

The system parameters for the simulation are:

·         Experimentation area (m2): 50.

·         Transmission range (m): 15-18.

·         Dimension between nodes (m): 10.

·         Packet rate (kbps): 250.

·         Protocol: ZTR.

·         Packet size (bytes): 100.

·         Number of coordinator: 1.

·         Number of attackers: 1.

·         Number of available channels: 11.

·         Experimentation time (sec): 100.

Table 8. Comparison of the algorithms: presented in this work and the cited in [19].

Table 8 shows a great similarity between performance metrics, taking into account that originally our proposed algorithm is made for a scenario of static nodes. However, the connected mechanism proposal shows very good performance in terms of overhead (approximately 17% less overhead than the other two proposals) and the jamming duration (approximately 18% less duration than the other two proposals). Likewise, the proposal of the extended mechanism has a performance lower than the algorithm cited in [19] of approximately 6%. Regarding energy consumption metric, we measure in both cases the total network consumption during the experimentation time. The best scenario that our work presents is the Extended mechanism, which has a consumption of 7% less energy with respect to the algorithm evaluated in [19].

Comment 2: The paper is not well written and needs to be proofread. The current version contains a lot of typos.

Response: Thank you very much. This is a pertinent comment and we have modified the text of the paper accordingly. We have corrected the manuscript in order to improve English and avoid excessive repetition of words.

Comment 3: The paper needs also to be reorganised. For example in the related work section, the authors explain some parts of their proposal, and come back again to other related work. It should be done separately, unless using the same behaviour of a part in a described work.

Response: Thank you very much for your recommendation and we rewrote the Related Work section in line with the reviewer's comment. You can observe these changes in the section because we added a subsection named “Proposed method and contributions”:

Our work focuses on finding an alternative where the network can defend itself, without the need of techniques that use large amount of resources, such as cryptography. In our analysis, we emphasize the study of one of the most aggressive cases of jamming (i.e., the reactive), and we propose two techniques inherent to the network so that the nodes, taking advantage of the communication capabilities of a sensor network, can defend against an attacker and counteract its effect. Our proposal incorporate the information of the jammed nodes to the tables of the neighbor nodes so that the self-configuring mechanisms of the routing protocols avoid the jammed area. Moreover, in this work we also present an algorithm for measuring thresholds of performance metrics to note the level of affectation of a node. We intend to cover the best of the previously studied proposals in order to test protocols widely known in sensor networks (proactive and reactive protocols) by modifying their valid routes so that it can guarantee a reliable delivery of information and isolate a possible attacking node from the other nodes in the network. In other words, it makes an accurate detection of the affected area based on more reliable performance metrics. Specifically, the contribution of this work to the state of the art are as follows: (1) We study a reactive jamming that represents the worst-case scenario and the most aggressive attack. (2) We also present an algorithm for measuring thresholds of performance metrics to note the level of affectation of a node. (3) We add a parameter in the routing tables and a control packet for AODV and DSR protocols. (4) Different from other works that find the best route to a destination, in this work we isolate the affected area so that the protocol re-configures the valid routes and the information arrives reliably.

Comment 4: Why the authors use their own simulator rather than the known ones like tossim and contiki, it is better to motivate this choice in few words.

Response: This is a pertinent recommendation and we have added a complete justification for using our tested simulator, as follows:

WSNs are extensively studied with several network simulators that analyze various performance and energy consumption metrics. However, the use of these simulators is directed to study some topologies or already defined and parametrized environments, as Network Simulator 2 (NS-2) [32]. We can also find network simulators as TOSSIM [33], that estimates the energy consumption while considering the batteries lifetime of the devices and set realistic scenarios with known platforms. This work enables us to build a network simulator based on an event-driven system where we have an approach to the Physical, MAC and, Network layers. Thus, we can simulate any topology, implementing several routing protocols, observe and count collisions, re-transmissions, channel retries and establish models to evaluate energy consumption techniques. Currently available simulators must have the possibility to allow flexibility for modifications to incorporate new protocols. Therefore, we designed and implemented a network simulator 280 based on events using the C++ language. The simulator was conceived with the paradigm of object-oriented programming (OOP), where nodes are autonomous entities (objects) that have properties and functions as transmit, receive and route packets. Simulation events are managed by a planner which serves as a "task organizer" for objects involved in the simulation. Some advantages of the C++ simulator are the speed of the program execution and the OOP that provides easiness of managing various classes as a separate entities that interact autonomously, for example, a node. We use this event-driven simulator based on C++ language proven in [34]. This is a network simulator with parameters of Physical, MAC and Network layers.

Comment 5: The tables and Figures are not well referenced, in many occurrences there is ??.

Response: Thank you very much. We have reviewed all the tables and figures and corrected them.

Comment 6: The algorithm 1 is not well written.

Response: This is also an appropriate comment and we have rewrite the algorithm in order to reflect the both mechanisms for jamming detection, as follows:

Comment 7: The table 2 should be explained better as the explanation is not sufficient. The same remark concerns all the figures and tables results.

Response: Thank you very much. We made a better explanation with all tables and figures along the manuscript.

Table 2:

Table 2 shows the percentage accuracy of each scenario with respect to the attack zone (where the jammer node should be found). In addition, it presents the percentage of false positives that are nodes incorrectly marked as attacked, but they should not be. This can be due to interference or some problems in the communication channel, but it diverts the attention of the network in the affected zone. The affected area represents the number of nodes that present performance metrics with anomalous values. Remember that the measured parameters for nodes are: energy, re-transmissions and retries for listening to the channel (CSMA retries). Under any of the two jamming detection mechanisms (Connected or Extended) the nodes compare their metric values, if with any of the considerations of each scheme these values are not similar, this node is marked and this type of marked nodes make up the affected area. However, in each detection mechanism, these decisions to mark nodes can fail and these nodes that are really affected become false positives. False positives make the affected area confused and the position of the jammer node is harder to find. In the simulation, we know where the jammer node is, but we need each detection mechanism to identify it completely. For this reason, we have chosen 3 positions to locate the jammer node in Figure 1: near to the coordinator node, middle in the topology, and far from the coordinator node. In each of the scenarios we know the number of nodes that surround the jammer node (according to the position it has in the topology), then we can know that for each position the detection mechanism must recognize the affected nodes. Then, the best scenario is the number 3 because the affected nodes are recognized at 100% in the three cases of position of the jammer node and the amount of false positives is much lower than in the other two scenarios (8%, 4%, and 4%). This means that the affected area is delimited with better precision and the affected nodes are clearly recognized.

Figure 2:

Figure 2 shows the results of the proposed Extended mechanism for identification of jammed nodes for the MPH protocol. The three possible positions of the jammer node are with respect to the collector node and are: near (position [2,2] of the grid matrix of the Figure 2), middle (position [4,4] in the Figure 2) and far (position [7,7] of the grid matrix of Figure 2). These positions are based on Figure 1. The results show in white squares the nodes that remain in the routing tables of their neighbors and in black squares those that are not included as neighbors, while the gray colors represent intermediate conditions (may be also under attack). This Figure has four columns representing four situations in the network: 1) No jamming or normal network conditions, 2) When the jammer node is close to the coordinator, 3) When the jammer node is in the middle of the topology, 4) When the jammer node is in the farthest corner from the coordinator node. The column "No jamming" is represented by the nodes in stable state or normal conditions, therefore, the nodes that are grayer indicate the nodes near to the collector node because they constitute a bottleneck in the network. The nodes that remain white are those that do not have their altered performance parameters. When we obtain the performance parameters in the other cases (near, middle and far) with the jammer node, a difference is established and when the difference is very low, almost zero, the squares are clearer (almost whites) and this indicates that they are remaining under their normal operating conditions. Then, when the difference in the performance parameters (energy, re-transmissions and CSMA retries) is high, in the "near", "middle" and "far" columns, these nodes become darker. The above represents that, in the last three columns, the darker nodes show the affected nodes according to each of the positions of the jammer node. This differential makes the "near", "middle" and "far" columns are relative, not absolute, values of the evaluation of the three variables on average: energy, re-transmissions and CSMA retries. In this Figure the scale is up to 3 because an average estimate of each metric is made and its value is weighted up to 3 so that when we introduce the jammer node, differences are found and hence the colors of the squares come out (which represent the nodes in the network) and we can identify affected nodes. Moreover, an area that is slightly affected in all cases is the proximity to the coordinator node because this becomes a bottleneck because in this type of network all the nodes send packets to the coordinator node, which is located in the lower left corner and, being a square topology, there are few nodes with direct access to this sink node.

Figure 3:

This simulations were run assuming Scenario 3, that is the scenario with better performance. The results show that we have about 10% better resolution of the affected area under jamming with the extended group approach. In this Figure, we look for the delimitation of the area where the jammer node is located. There are three positions of the jammer node (near, middle and far) and the darkest nodes are those that are detecting the affected area. The comparisons among performance metrics are done with more information in the Extended mechanism case, which provides a more assertive decision for sending the control packet. In the Connected approach, the information is only gathered from the directly connected neighbors, so the self-aware algorithm is less efficient. Both mechanisms collect information on node performance metrics that have the same characteristics with respect to proximity to the coordinator node.

Figure 4:

We also run a simulation under network stress conditions. With this type of simulation we want to observe the behavior of both mechanisms in the best simulation scenario (Scenario 3), but in the worst case. One of the cases is a high packets loss in the network due to interference in the links or connections and disconnections of nodes. The other case is due to high traffic of information packets, which increases the overhead along the network and therefore, packet collisions. The previous cases cause that more packets are retransmitted and the channel is constantly busy, so the CSMA retries also increase. In Figure 4, the first case is represented by packet loss (between 5 and 10\% per link) in the links due to possible interference in the channel. Thus, links acquire simulated loss percentages so that we measure if the attack made by the jammer node is camouflaged or not. We observe that the Extended mechanism has a behavior of around 6% better resolution of the affected area with respect to the Connected mechanism. In both approaches, it is possible to visualize the area in the presence of the jammer node, however, the Extended algorithm has a better definition.

Figure 5:

In Figure 5 network stress is achieved by sending high amount of traffic packets to the collector node (20 packets per second per node). In this figure we also seek to stress the network in order to observe if the attack of the jammer node is camouflaged due to the increase of values in the performance parameters thanks to the overhead that is being imposed on the network. The difference in the resolution of the area between the two studied mechanisms is 8%, with the Extended mechanism being of better resolution with respect to the Connected mechanism.

These two figures show the simulations where the network is under high stress conditions and a possible attack could be camouflaged in this situation. Results show that in the presence of packet loss or high network traffic, considering Scenario 3 which is the most complete because it has detection and mitigation, the Extended mechanism presents a resolution of the affected area 7\% greater than in the Connected mechanism. Another positive aspect of this analysis is that both proposed mechanisms in this work allow the detection of the affected area, where possibly, the jammer node will be under extreme conditions. The accuracy percentage of affected nodes correctly identified is 75% for the Extended mechanism and 68\% for the Connected mechanism.

Table 4:

Table 4 shows the energy consumption for each of the node's main activities at network level. \hl{We make this comparison to base the consumption on the energy model proposed for the sensors, detailed in Table 3. Each of the energies described here are total energies for 30 seconds of simulation time and based on the grid described in Figure 1. We expose the two techniques of jamming detection when the jammer node is in the middle of the topology, jamming is reactive, and the percentage of packet loss per link is between 0.5 and 2.5%. We note that the Extended Mechanism improves energy consumption by about 20% against Connected Mechanism. This table checks the energy consumption of the nodes under the reactive jamming, which is the most aggressive of all the cases for the scenarios we have proposed. We analyze each one of the energy types of the model to visualize where the greatest energy expenditure is made at the moment of a network under attack conditions. The types of energy that have the greatest impact are: the energy of the CSMA algorithm, the receiving energy and the transmission energy. This situation is logical because, under attack, the amount of overhead in the network increases and, therefore, increase packet re-transmissions, retries to listen to the communication channel and collisions.

Figure 6:

Figure 6 presents the shaded area for the three performance metrics the Scenario 3. Thanks to the simulation results obtained so far, it is important to scale the model to have a reproduction of a medium network and be able to analyze the behavior of each of the performance metrics: re-transmissions, CSMA retries and energy. After careful observation and analysis of a matrix of 225 nodes we noticed that the method of detection and mitigation used can be extrapolated to a greater number of nodes. We expose the Extended technique for jamming detection when the jammer node is in the middle of the topology, jamming is reactive, and the percentage of packet loss per link is between 0.5 and 2.5%. The simulations are run for three characteristic protocols that represent different approaches to know routes in the network: reactive protocols (AODV and DSR), proactive protocol (MPH). In this way, we have a broader analysis of the Extended mechanism's impact on different communication protocols and a greater number of nodes. The darker area shows the possibility of finding the jammer node there. Note that in the proactive protocol, such as MPH, the delimitation of that area is reduced to 10% of the network, whereas in reactive protocols, such as AODV and DSR, it is reduced to 25% of the complete topology.

Figure 7:

Figure 7 shows the energy per node when the network is under normal conditions and when the network is under attack and the attacking node is in the middle of the topology (Position 24 in Figure 1). Remember that we have 49 nodes in a square topology and we want to know the total energy (in Joules) per node. When the network is under normal (stable) conditions, nodes 1, 7 and 8 present a normal increase in energy consumption because they are the closest nodes to the coordinator node and, because of the traffic distribution packets of the rest of the nodes in the network. Now, when the network is under jamming, the nodes that have the greatest impact on energy consumption should be 16, 17, 18, 23, 25, 30, 31 and 32, because it is the area of those directly connected with the jammer node. Indeed, this happens for the three studied protocols and the peaks that are observed in the graph just the levels corresponding to these mentioned nodes, including the jammer node, because this node is running under a reactive jamming where the energy consumption is higher. As shown, the total average energy for AODV is 0.1209, for DSR is 0.1105 and for MPH is 0.0397. Therefore, the MPH protocol is more energy efficient by 74% than AODV, and 69% than DSR considering the average energy of the nodes.

Figure 8:

In Figure 8, we compare the performance in terms of the persistence parameter of MPH and two others widely known protocols, AODV and DSR. Remember that the persistence parameter, p, indicates the soft exit of the nodes from the neighbor tables of the other nodes, depending on the scenario (previously we mentioned Scenario1, 2 and 3). This exit of nodes from the tables is due to the distrust that this node has because its performance metrics are having anomalous values. Another point worth noting is that the Figure 8 describes the location of the jammer node for three different positions (near, middle and far from the collector node). Simulations were performed considering Scenario 3 and modifying AODV and DSR protocols by including the transmission of a control packet. This control packet is sent in Scenario 3 to inform a node that it must remove another from its neighbor table, in order to no longer have that route as a path in the network. The nodes ejected through this control packet seek to isolate the possible jammer node without it notices and, for this reason, will continue to carry out its attack. The disadvantage that could arise in this scheme is some network routes are lost, because we disconnect some nodes (the nodes directly connected to the possible jammer node), however, the mitigation of the attack will not impact on the other nodes and the energy of the jammer node can be consumed more quickly. The results show that AODV and DSR are capable of reducing the area of the location of the jammer node to approximately 15% of all the network while MPH can further reduce the area to 4%. This ensures that the attacker node may be better located under a possible jamming with a reduced zone. The smaller the area, the more precise the location of the jammer node and the fewer nodes in the network will be disconnected.

Table 5:

Table 5 complements the analysis of Figure 8 under the Scenario 3. This Table has a simulation time of 100 seconds and analyzes the reactive jamming of a jammer node located in the middle of the topology. The Table analyzes different performance metrics such as: re-transmissions, CSMA retries, end-to-end delay, resilience, affected nodes, and energy. Re-transmissions are the times that packets that are not received and they are sent again. We observe that the MPH protocol has a 28% lower re-transmissions with respect to AODV and DSR. For CSMA retries, MPH has 18% fewer attempts to listen to the channel than AODV and DSR, under a state of a jamming reactive. Regarding end-to-end delay, packets under the MPH protocol arrive at the coordinator node 0.51 seconds faster than packets in AODV and DSR. These three metrics indicate that the MPH protocol has better reactivity to isolate the attack of a jammer node against AODV and DSR. Valid routes of MPH are faster than those of AODV and DSR. However, AODV is still that presents the metrics with the lowest performance. The resilience is the ability of the network to recover from an abnormal state, in this case we run the simulation for the same 100 seconds and the jammer node is activated in the second 30, for 10 more seconds and then disappears. We want to observe how the network is recovered under each of the protocols and we note that MPH has better resilience in 18% with respect to AODV and DSR. The metric of affected nodes is measured in nodes that change their values of metrics: energy, re-transmissions and CSMA retries, in an anomalous way. This metric is closely related to false positives, because they are collateral effects that are generated according to each protocol under jamming. Finally, the MPH protocol has a lower energy consumption of 58% with respect to AODV and DSR. This represents that MPH has a very good recovery reactivity when the network is under attack, specifically, an attack in a focused area at almost the same distance from all nodes in the network (including the coordinator node).

Figure 9:

Figure 9 shows a direct comparison between the three protocols under the two proposed jamming techniques, based on the total energy for each of the three protocols. This Figure is very interesting because it shows the energy performance of each node in each of the three protocols studied. The simulation was made for 100 seconds, under Scenario 3 and the jammer node is located in the middle of the topology emitting a reactive jamming. Again, we note that nodes 1, 7 and 8 are the nodes that forward all network traffic to the coordinator node, and because of the topology, these nodes have a high traffic load and their performance metric values increase more than in the other nodes in the network. Nodes 16, 17, 18, 23, 25, 30, 31 and 32 are those that directly limit the jammer node. These nodes present different behavior than the rest of the nodes of the network and, therefore, their energy increases in a noticeable way. First of all, it is easy to note how MPH has a better performance than DSR and AODV with a value of 0.0390 on the extended group and 0.0306 on the connected group since DSR has a value of 0.0758 and 0.1062 and AODV has a value of 0.0984 and 0.1147 on the extended group and connected group respectively.

Figure 10:

In Figure 10we represent the energy values for each of the three protocols in a box and whiskers scheme. A graph of this type consists of a rectangular box, where the longer sides show the interquartile path. A vertical segment that indicates where the median is positioned and therefore its relation to the first and third quartiles divides this rectangle. This box is located on a scale on a segment that has as ends the minimum and maximum values of the variable. The lines that protrude from the box are called whiskers. These whiskers have a prolongation limit, so any data or case that is not within this range is marked and identified individually. On the bottom of the box is shorter than the one on the right in the three studied protocols; so 25% of the nodes that spend less energy are more concentrated than 25% of those which consume more. In addition, for the three protocols, the energy consumed by nodes between 25% and 50% of the population is more dispersed than between 50% and 75%. We observe that the displacement of the box plots downwards, especially in the MPH protocol, indicating that here the majority of nodes spend less energy in both mechanisms, being the difference between the maximum and the lowest minimum, as well as the interquartile difference. The results show that in the AODV protocol the energy distribution is more dispersed than in the other two protocols and there is a higher energy expenditure compared to the other two protocols. For the MPH protocol, there is more atypical data than in the other two protocols because in both mechanisms, the nodes that surround the jammer node consume more energy, a parameter that is very noticeable in order to isolate an area of the network.

Table 6:

The Table 6 shows values of performance metrics along the network. \hl{This Table describes a comparison of the three protocols studied with the modifications made in each detection mechanism jamming (Extended and Connected). The tests were made with non-ideal links and traffic rate of 10 packets per second per node under the grid topology shown in Figure 1. Metrics are average values of re-transmissions and CSMA retries. The delay end-to-end is calculated as the average per route. Overhead is a metric that influences the amount of network collisions and the channel occupancy. Overhead indicates the number of control packets under a routing protocol. It depends on the number of routing packets that the routing protocol needs to connect nodes and to route packet traffic. Thus, it is calculated taking into account the number of control packets that are needed to route a traffic packet. Ideally, a routing protocol should need the least amount of control packets. Metric such as Energy is given by the average energy per node along the network.} Resilience is a parameter that measures the recovery time of a system. This metric is analyzed under the network under normal conditions and after 30 seconds the jammer node is introduced in the middle of the topology and after 10 seconds the jammer node goes out and here we observe and measure how long it takes the network to recover and return to its normal conditions. The affected nodes is a metric that is set as a ratio of nodes that appear altered in their performance parameters and are those that are actually encircling the jammer node between nodes that ideally should surround the jammer node completely. For this grid theoretically establish 9 nodes that should completely surround if the jammer node is in position 24 of the topology in Figure 1.

Comment 8: Why the scenario with no jamming in figure 3 is black (and also other figures). It should not be white for the cause of absence of jammers? Perhaps I don’t understand well the mechanism, for that I propose more explanations. In general, for the different figures I don’t understand the relation between the no jamming scenario and the black boxes . I think that without jamming, it should be white, not?

Response: Thank you very much and you are right. We made a better explanation with this topic and we modified the figures in order to clarify the part of “No jamming”, as follow since the Figure 2:

Figure 2 shows the results of the proposed Extended mechanism for identification of jammed nodes for the MPH protocol. The three possible positions of the jammer node are with respect to the collector node and are: near (position [2,2] of the grid matrix of the Figure 2), middle (position [4,4] in the Figure 2) and far (position [7,7] of the grid matrix of Figure 2). These positions are based on Figure 1. The results show in white squares the nodes that remain in the routing tables of their neighbors and in black squares those that are not included as neighbors, while the gray colors represent intermediate conditions (may be also under attack). This Figure has four columns representing four situations in the network: 1) No jamming or normal network conditions, 2) When the jammer node is close to the coordinator, 3) When the jammer node is in the middle of the topology, 4) When the jammer node is in the farthest corner from the coordinator node. The column "No jamming" is represented by the nodes in stable state or normal conditions, therefore, the nodes that are grayer indicate the nodes near to the collector node because they constitute a bottleneck in the network. The nodes that remain white are those that do not have their altered performance parameters. When we obtain the performance parameters in the other cases (near, middle and far) with the jammer node, a difference is established and when the difference is very low, almost zero, the squares are clearer (almost whites) and this indicates that they are remaining under their normal operating conditions. Then, when the difference in the performance parameters (energy, re-transmissions and CSMA retries) is high, in the "near", "middle" and "far" columns, these nodes become darker. The above represents that, in the last three columns, the darker nodes show the affected nodes according to each of the positions of the jammer node. This differential makes the "near", "middle" and "far" columns are relative, not absolute, values of the evaluation of the three variables on average: energy, re-transmissions and CSMA retries. In this Figure the scale is up to 3 because an average estimate of each metric is made and its value is weighted up to 3 so that when we introduce the jammer node, differences are found and hence the colors of the squares come out (which represent the nodes in the network) and we can identify affected nodes. Moreover, an area that is slightly affected in all cases is the proximity to the coordinator node because this becomes a bottleneck because in this type of network all the nodes send packets to the coordinator node, which is located in the lower left corner and, being a square topology, there are few nodes with direct access to this sink node.

Comment 9: The caption of figure 7 does not match with the explanation of section 4.3

Response: We thank the reviewer for his/her comment. We corrected the Figure 7, the new caption is “Comparison of energy per node under stable and attacked conditions”.

Comment 10: Generally, all the figures need more analysis to explain the results, as it is not well done.

Response: We thank the reviewer for his/her comment. We have explained in detail and in greater depth all the tables and figures.

Comment 11: The bibliographic references need also to be more organised. Some are duplicated like 12 and 16.

Response: Thank you very much. We have corrected the references as you indicated.

Comment 12: Do the authors evaluate the consequences of isolating the affected routes in case there is no other routes, rather than choosing the best of the existing ones (the less affected)

Response: Thank you very much. We have added an explanation about this concern with the Table 5, as follow:

The advantage that Scenario 3 presents, which is the best scenario chosen and tested in most simulations, is that routes of possibly affected nodes are not eliminated immediately, but that the nodes present a smooth exit from the routing tables of other nodes. These consequences of route isolation are analyzed when we study the resilience in each of the protocols. The recovery capacity of the network shows how you can get to re-establish routes that had been isolated and how it impacts the fact of isolating them in the performance of the network. Then, the proposed detection mechanisms (Connected and Extended) do not completely eliminate the affected nodes (or what is the same, the affected routes), but it locates these routes as less secure. The above is clearly observed in that none of the protocols completely loses their route redundancy (for example, we have a small network of 49 nodes) and the resilience capacity of the network in none of the protocols decreases by 75%.

Thank you very much.

Sincerely,

Carolina Del Valle Soto

Universidad Panamericana. Facultad de Ingeniería. Álvaro del Portillo 49, Zapopan, Jalisco, 45010, México.

Phone: +52 (33) 13682200 | Ext. 4245

Reviewer 2 Report

The authors propose two methods called the Connected Mechanism and the Extended Mechanism for detecting the jamming in the WSNs. After reading this paper, I have some comments to the authors:

The research subject is interesting. But, the scenario of the WSN environment is limited in the discussion. It will be better if the authors can make an estimation on how their method will perform when the connection and/or the deployment of WSNs are based on some specific patterns such as energy saving.

Some indices are missing in the paper. For example, you have reference problems in line 293 and line 311 shown as "in Table ??"

The experiments are performed on the platform implemented in C++ by the authors. I'm not sure whether it is an open platform for others to access, but at least all parameters and assumptions of the experiments should be revealed for the reader to reproduce the experimental results.

Referring to comment 1, I suggest the authors include references listed as follows and others in the related research topic to have a comprehensive literature review:
i) Jeng-Shyang Pan, Lingping Kong, Tien-Wen Sung, Pei-Wei Tsai and Waclav Snasel, “α-Fraction First Strategy for Hirarchical Wireless Sensor Neteorks”, Journal of Internet Technology, Vol. 19, No. 6, pp. 1717~1726, 2018. --> (Energy-saving oriented scenario)
ii) Jing Zhang, Shi-Jian Liu, Pei-Wei Tsai, Fu-Ming Zhou, and Xiao-Rong Ji, May 2018, “Directional Virtual Backbone based Data Aggregation Scheme for Wireless Visual Sensor Networks,” PLoS One, DOI: 10.1371/journal.pone.0196705. --> (Transmission path predefined scenario)
iii) Yung-Fa Huang and Chung-Hsin Hsu, "Energy Efficiency of Dynamically Distributed Clustering Routing for Naturally Scattering Wireless Sensor Networks," Journal of Network Intelligence, Vol. 3, No. 1, pp. 50-57, Feb 2018. --> (Dynamic routing scenario)
If the authors can convince readers that the proposed methods can be applied in different WSN scenarios, then the contribution of this work can be further extended.

Author Response

Dear

Editor

Sensors Editorial Office

We are submitting the paper:

“New Detection Paradigms to Improve Wireless Sensor Network Performance under Jamming Attacks”

Authored by: Carolina Del-Valle-Soto, Carlos Mex-Perera, Ivan Aldaya, Fernando Lezama, Juan Arturo Nolazco-Flores and Raul Monroy

We would like to thank the reviewers and editors for their detailed analysis of the manuscript; the comments are very valuable to us. In the revised version of the paper we have incorporated the changes recommended by the reviewers.

Comments to all observations and suggestions including point-by-point responses are addressed in the following text.

_______________________________________________________________________________

Reviewer 2 comments

Comment 1: The authors propose two methods called the Connected Mechanism and the Extended Mechanism for detecting the jamming in the WSNs. After reading this paper, I have some comments to the authors:

The research subject is interesting. But, the scenario of the WSN environment is limited in the discussion. It will be better if the authors can make an estimation on how their method will perform when the connection and/or the deployment of WSNs are based on some specific patterns such as energy saving.

Response: We thank the reviewer for his/her observations and comments that have contributed to the improvement of the paper. We added a subsection named “Comparison with Another Jamming Detection Algorithm”. In this section, we compare our jamming detection mechanisms with another mechanism of the literature and analyze a new topology under real experimentation with wireless sensors CC2650 of Texas Instruments. We analyze different metrics under a deployment based on patterns such as: overhead, PDR and energy.

We added the following paragraph in the Related Work section in order to introduce the algorithm to compare our detection mechanisms in other section:

Jamming is considered one of the most harmful attacks in threats to WSNs. A jamming attack can limit the communication capabilities of a WSN by degrading the system performance interfering with signals using certain interfering devices [18]. For instance, the proposal in [19] is based on a jamming detection algorithm that employs honeynodes and response mechanism concepts. Each node cooperates with the transmitter to detect any jamming. The authors analyze network performance metrics such as: duration of jamming attack, control message overhead, and the number of channel re-configurations, as well as One of the metrics they use in their simulations is the Packet Delivery Ratio (PDR).

Moreover, we added a subsection called “Comparison with Another Jamming Detection Algorithm”, as follow:

In order to compare our proposal with the algorithm cited in [19], we will explain better the mechanism mentioned in this work as a jamming detection method. When any of the mobile nodes or base-stations detect an attack, the node changes its frequency based on a pseudorandom sequence. If no attack is detected, it checks if the node is a base station or if it is a real node. One of the similar characteristics with our algorithm is that in both the associated nodes are informed about changing their state. In the algorithm quoted in [19] the reported nodes change their frequency of operation. In the mechanism proposed in this paper (connected and extended algorithms), the informed nodes change their parameters to make a detour of the affected area. Continuing with the algorithm cited in [19], in the case that the node is a base station when a node has not been informed of the attack, beacons are sent to the associated nodes, and the nodes that have not responded are searched. In the case where the node is an authentic node, if it received the change of frequency by the access point, its frequency changes. Otherwise, it listens for beacons. Then, if the beacons are not received, the node changes its frequency of operation, otherwise, it responds to the beacon.

Table 8. Contributions of each algorithm

We have a scenario that consists of 5 nodes located in a star topology and a coordinator (6 nodes in total) in order to test the comparison of algorithms. This arrangement is described in Figure 11. In order to create a ZigBee network it is necessary to use the CC2531 as a coordinator because the CC2650's flash is not enough to build the network. The CC2650 LaunchPad includes: External 8Mbit Serial Flash for supporting Over the Air (OAD) firmware updates, 2x push buttons, complete contents of CC2650EM-7ID, the evaluation module, 2x LEDs, 2x BoosterPack connectors, standardized LaunchPad form factor XDS110 debugger with external target interface. We use the 802.15.4 protocol. Easy link is a protocol for transmitting packets and creating networks through the 2.4Ghz frequencies. It is also possible to use the CC2650 as a sniffer, to acquire the ZigBee packages. In addition, a CC2530 node from Texas Instruments has been used as a coordinating device. The evaluation board used to program it is the SmartRF05 evaluation board.

Figure 11. Network configuration for comparison of algorithms.

The system parameters for the simulation are:

·         Experimentation area (m2): 50.

·         Transmission range (m): 15-18.

·         Dimension between nodes (m): 10.

·         Packet rate (kbps): 250.

·         Protocol: ZTR.

·         Packet size (bytes): 100.

·         Number of coordinator: 1.

·         Number of attackers: 1.

·         Number of available channels: 11.

·         Experimentation time (sec): 100.

Table 8. Comparison of the algorithms: presented in this work and the cited in [19].

Table 8 shows a great similarity between performance metrics, taking into account that originally our proposed algorithm is made for a scenario of static nodes. However, the connected mechanism proposal shows very good performance in terms of overhead (approximately 17% less overhead than the other two proposals) and the jamming duration (approximately 18% less duration than the other two proposals). Likewise, the proposal of the extended mechanism has a performance lower than the algorithm cited in [19] of approximately 6%. Regarding energy consumption metric, we measure in both cases the total network consumption during the experimentation time. The best scenario that our work presents is the Extended mechanism, which has a consumption of 7% less energy with respect to the algorithm evaluated in [19].

Comment 2: Some indices are missing in the paper. For example, you have reference problems in line 293 and line 311 shown as "in Table ??"

Response: Thank you very much. We have reviewed all the tables and figures and corrected them.

Comment 3: The experiments are performed on the platform implemented in C++ by the authors. I'm not sure whether it is an open platform for others to access, but at least all parameters and assumptions of the experiments should be revealed for the reader to reproduce the experimental results.

Response: This is a pertinent recommendation and we have added a complete justification for using our tested simulator, as follows:

WSNs are extensively studied with several network simulators that analyze various performance and energy consumption metrics. However, the use of these simulators aims at the analysis of some topologies or already defined and parametrized environments, as Network Simulator 2 (NS-2) [28]. We can also find network simulators as TOSSIM [29], that estimates the energy consumption while considering the batteries lifetime of the devices and set realistic scenarios with known platforms. This work enables us to build a network simulator based on an event-driven system where we have an approach to the Physical, MAC and, Network layers. Thus, we can simulate any topology, implementing several routing protocols, observe and count collisions, re-transmissions, channel retries and establish models to evaluate energy consumption techniques. Currently available simulators must have the possibility to allow flexibility for modifications to incorporate new protocols. Therefore, we designed and implemented a network simulator based on events using the C++ language. The simulator was conceived with the paradigm of “object-oriented programming” (OOP), where nodes are autonomous entities (objects) that have properties and functions as transmit, receive and route packets. Simulation events are managed by a planner who serves as a ”task organizer” for objects involved in the simulation. Some advantages of having made the simulator in C++ is the speed and OOP provides easiness of managing various classes as a separate entity that interacts autonomously, for example, a node.

We use this event-driven simulator based on C++ language reported in [32]. This is a network simulator with parameters of Physical, MAC and Network layers.

Moreover, Table 1 specifies the simulation parameters for the described simulator.

Comment 4: Referring to comment 1, I suggest the authors include references listed as follows and others in the related research topic to have a comprehensive literature review:

i) Jeng-Shyang Pan, Lingping Kong, Tien-Wen Sung, Pei-Wei Tsai and Waclav Snasel, “α-Fraction First Strategy for Hirarchical Wireless Sensor Neteorks”, Journal of Internet Technology, Vol. 19, No. 6, pp. 1717~1726, 2018. --> (Energy-saving oriented scenario)

ii) Jing Zhang, Shi-Jian Liu, Pei-Wei Tsai, Fu-Ming Zhou, and Xiao-Rong Ji, May 2018, “Directional Virtual Backbone based Data Aggregation Scheme for Wireless Visual Sensor Networks,” PLoS One, DOI: 10.1371/journal.pone.0196705. --> (Transmission path predefined scenario)

iii) Yung-Fa Huang and Chung-Hsin Hsu, "Energy Efficiency of Dynamically Distributed Clustering Routing for Naturally Scattering Wireless Sensor Networks," Journal of Network Intelligence, Vol. 3, No. 1, pp. 50-57, Feb 2018. --> (Dynamic routing scenario)

If the authors can convince readers that the proposed methods can be applied in different WSN scenarios, then the contribution of this work can be further extended.

Response: Thank you very much for your recommendation and we added these references to our Related Work. You can observe these changes in the section.

In [23], Pan et al. study problems related to energy depletion in hierarchical WSNs models. The authors propose an energy-saving oriented model in which a relay node is used to regulate packet traffic and energy consumption in intermediate nodes. Data gathering, another issue presented in hierarchical networks, is analyzed in [24] under a predefined transmission path scenario. Since a predefined transmission path favors the creation of clusters, a dynamic routing approach is proposed in [25] to balance the traffic in the network and decrease the energy consumption in the nodes. A new performance metric, called energy consumption density, is established to validate the effectiveness of their approach. Against that background, the proposed jamming detection technique is implemented in a hierarchical model of WSNs, which facilitates packet sending and decreases network overhead. Besides, most of the studies use energy-related metrics to analyze the performance of long-lasting and difficult access networks such as wireless sensor networks. In contrast, we also introduce five performance metrics to enhance the accuracy in the detection of a possible attacker. The additional metrics proposed in this work are i) re-transmissions, ii) CSMA retries, iii) resilience, iv) delay and v) valid routes.

Thank you very much.

Sincerely,

Carolina Del Valle Soto

Universidad Panamericana. Facultad de Ingeniería. Álvaro del Portillo 49, Zapopan, Jalisco, 45010, México.

Phone: +52 (33) 13682200 | Ext. 4245

Round  2

Reviewer 2 Report

This is a revision of the previous submission. All my comments in the previous review have been answered by corresponding modifications in the article. I think this paper is ready to be accepted.